# Structural mapping of PEAK pseudokinase interactions identifies 14-3-3 as a molecular switch for PEAK3 signaling

Michael J. Roy [1,2] ✉, Minglyanna G. Surudoi [1,2], Ashleigh Kropp[1,2], Jianmei Hou[3,4], Weiwen Dai[1,2], Joshua M. Hardy [1,2], Lung-Yu Liang[1,2], Thomas R. Cotton [1,2], Bernhard C. Lechtenberg [1,2], Toby A. Dite[1,2], Xiuquan Ma [3,4], Roger J. Daly [3,4], Onisha Patel [1,2,5] & Isabelle S. Lucet [1,2,5] ✉

PEAK pseudokinases regulate cell migration, invasion and proliferation by recruiting key signaling proteins to the cytoskeleton. Despite lacking catalytic activity, alteration in their expression level is associated with several aggressive cancers. Here, we elucidate the molecular details of key PEAK signaling interactions with the adapter proteins CrkII and Grb2 and the scaffold protein 14-3-3. Our findings rationalize why the dimerization of PEAK proteins has a crucial function in signal transduction and provide biophysical and structural data to unravel binding specificity within the PEAK interactome. We identify a conserved high affinity 14-3-3 motif on PEAK3 and demonstrate its role as a molecular switch to regulate CrkII binding and signaling via Grb2. Together, our studies provide a detailed structural snapshot of PEAK interaction networks and further elucidate how PEAK proteins, especially PEAK3, act as dynamic scaffolds that exploit adapter proteins to control signal transduction in cell growth/motility and cancer.

The pseudopodium-enriched atypical kinase (PEAK) family of proteins, which comprises Sugen kinase 269 (Pseudopodium-enriched atypical kinase 1, PEAK1)[1], SgK223 (PEAK2), an ortholog of rat Pragmin[2] and mouse Notch activation complex kinase (NACK)[3], and the recently identified PEAK3 (C19orf35)[4], play critical roles in actin cytoskeleton remodeling, influencing cell migration and invasion in normal and cancer cells[5–8]. Abnormal expression of PEAK proteins alters cell morphology and confers enhanced migratory and invasive characteristics to cells[1,9–12], implying a role for PEAKs in the spatio- and temporal assembly of signaling hubs at focal adhesions (FAs) and in regulating the actin cytoskeleton[4,13–16].

PEAK proteins are a unique group of pseudokinase (PsK) scaffolds that can self-assemble regulatory networks. Their scaffolding activities rely on dimerization, post-translational modifications and recruitment of modular adapter proteins. PEAK proteins homo- and/or heterodimerize via a unique alpha-helical domain, the split helical dimerization (SHED) domain which flanks the PsK domain (Fig. 1), as recently elucidated by the structures of PEAK2[17,18] and PEAK1[19].

In addition to the conserved PsK and SHED domains, all PEAK proteins have an N-terminal intrinsically disordered region (IDR) (Fig. 1) replete with short linear motifs (SLiMs)−conserved sequence motifs that can that form dynamically regulated docking sites to recruit interactors involved in PEAK signaling. To date, these IDRs have only been partially characterized[4,14,17–20]. Several tyrosine phosphorylation sites have been identified within the large IDRs of PEAK1 and PEAK2 (IDR length of ~1200 and ~900 residues, respectively) that

[1]The Walter and Eliza Hall Institute of Medical Research, Parkville, VIC 3052, Australia. [2]Department of Medical Biology, University of Melbourne, Parkville, VIC 3052, Australia. [3]Cancer Program, Biomedicine Discovery Institute, Monash University, Melbourne, VIC 3800, Australia. [4]Department of Biochemistry and Molecular Biology, Monash University, Melbourne, VIC 3800, Australia. [5]These authors jointly supervised this work: Onisha Patel, Isabelle S. Lucet. ✉e-mail: roy@wehi.edu.au; lucet.i@wehi.edu.au

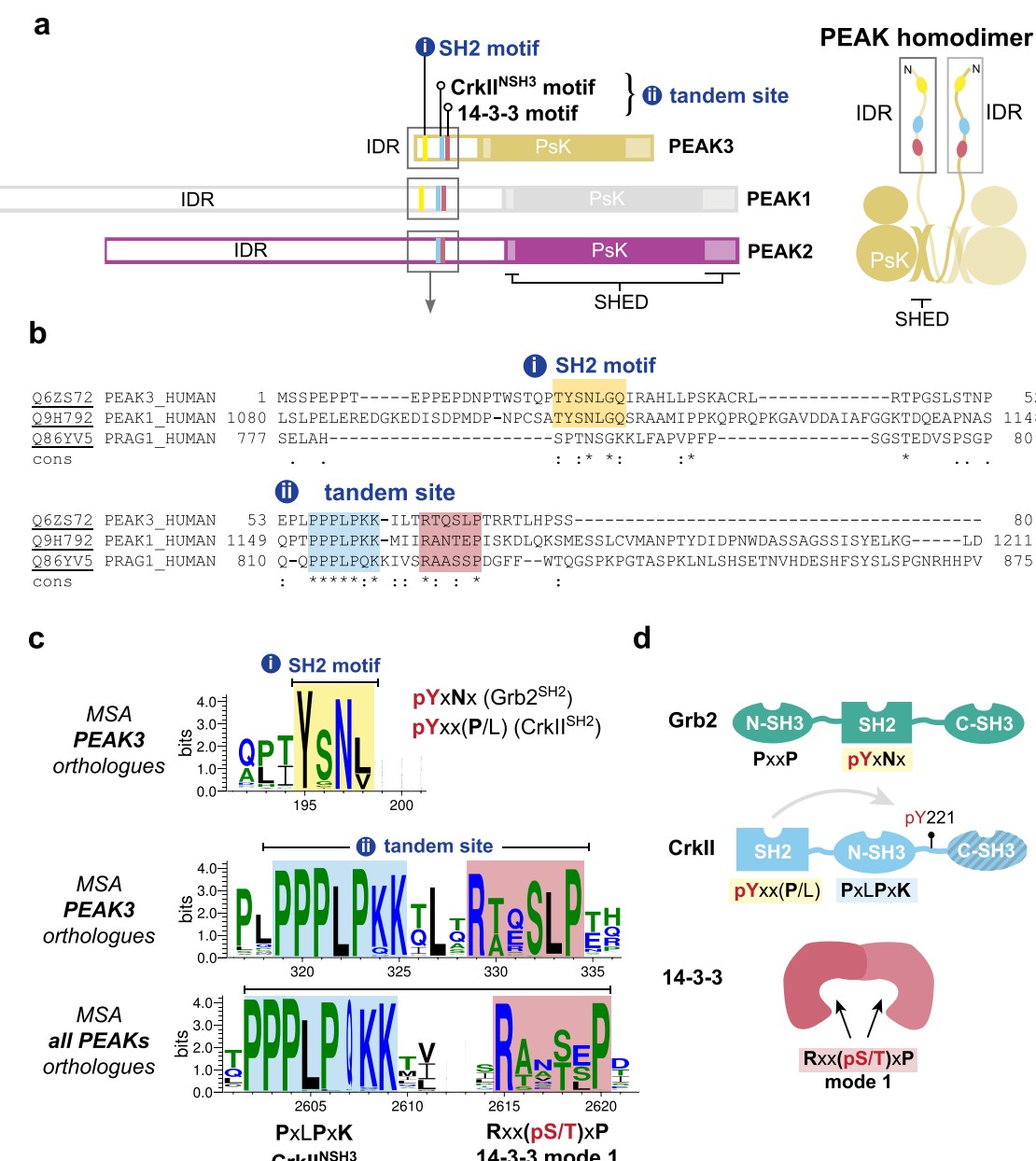

**Fig. 1 | PEAK domain organization and interaction motifs in N-terminal IDR.**
**a** Domain organization of the PEAK family and diagram of PEAK homodimer arrangement showing the SHED, PsK and IDR motifs identified (boxed). **b** Sequence alignment of the N-terminal IDR of human PEAK3 with the corresponding region of PEAK1 and PEAK2. **c** Multiple sequence alignment (MSA) of PEAK vertebrate orthologues highlighting short linear interaction motifs (SLiMs) identified in regions of high sequence conservation, including a pY/SH2 motif (Grb2[SH2]/CrkII[SH2]) and the tandem site encompassing a proline-rich motif (CrkII[NSH3]) and conserved putative 14-3-3 motif. **d** Schematic showing overall domain organization of Grb2, CrkII and 14-3-3, highlighting sequence motifs.

regulate cell proliferation, migration and invasion, and in some cases the recruited proteins with Src Homology 2 (SH2) and phosphotyrosine binding (PTB) domains have been characterized. In PEAK1, Tyr[665] is phosphorylated by Src, linking PEAK1 to the Src-p130Cas-Crk-paxillin pathway that regulates FA dynamics[21]. Additionally, phosphorylation of Tyr[635] by Lyn kinase creates a binding site for the SH2 domain of the adapter protein Grb2, which leads to the activation of the Ras/Raf/Erk signaling pathway that controls cell proliferation and invasion[9]. Both PEAK1 and PEAK2 are linked to the signaling adapter Shc1[7,22] and share a conserved "EPIYA" phosphotyrosine motif (PEAK2 Y[411]; PEAK1 Y[616]), which serves as a docking site for the SH2 domain of the C-terminal Src kinase (Csk). This interaction brings Csk to FAs where it likely regulates cell morphology and cell motility by modulating the activity of Src family kinases (SFKs)[16].

In contrast, the role of PEAK3 in signaling is poorly understood. PEAK3 has a more restricted expression profile than PEAK1/PEAK2; predominantly expressed in granulocytes and monocytes (Human Protein Atlas, proteinatlas.org)[23] suggesting a specialized function[8]. PEAK3's IDR is significantly smaller than PEAK1/PEAK2 (PEAK3 IDR length ~130 residues), but still harbors various SLiMs, some of which are conserved in PEAK1 and PEAK2. All reported cellular interactions of PEAK3 with binding partners appear to require PEAK3 dimerization[4,14,23]. Amongst PEAK3 interactors identified from proteomic/cellular studies are the adapter proteins CrkII/CrkL[24]. CrkII/CrkL are highly similar but non-identical proteins, with roles in focal adhesion signaling, cell migration, and cancer[24]. They both share the identical domain architecture (SH2-NSH3-CSH3) (Fig. 1d) and identical binding preferences in SH2 and NSH3 domains but are reported to

have a distinct structural architecture and interactome[24]. PEAK proteins have been shown to interact with CrkII via a conserved proline-rich motif (PRM) present in the N-terminal IDR of all PEAK proteins that binds the CrkII NSH3 domain (CrkII[NSH3]; consensus sequence PxLPxK) (Fig. 1d)[4,23]. Additionally, for PEAK3, and likely also PEAK1/2, CrkII binding requires PsK dimerization via the SHED domain and is impacted by mutations which disrupt the PsK domain conformation and dimerization[4,14]. Beyond this, two recent studies have identified further PEAK3 interactors, including: adapter protein Grb2; E3 ubiquitin ligase Cbl; proline-rich tyrosine kinase 2 (Pyk2); and Arf GTPase-activating protein 1 (ASAP1)[14,23]. Grb2 (domain organization NSH3-SH2-CSH3) is involved in multiple cellular signal transduction pathways, most prominently downstream of the epidermal growth factor (EGF) receptor to the mitogen-activated protein kinase (MAPK) signaling cascade (via Grb2/Sos complex)[25] but also in survival (phosphatidylinositol 3-kinase (PI3K)/AKT) signaling (via Grb2-associated binder 1, Gab1)[26]. Additional PEAK3 interactors identified include 14-3-3 proteins[4], a group of ubiquitous dimeric regulatory/scaffold proteins, of which there are seven isoforms in humans (β, γ, ε, η, σ, τ, and ζ). 14-3-3-proteins recognize specific phosphoserine/threonine motifs in partner proteins, including kinases, and often have regulatory roles in cellular pathways, such as to alter conformation, activity or subcellular localization[27–29].

In this study, we use an integrated bioinformatic, biochemical, and structural biology approach to further characterize the interactions and scaffolding activity of PEAK proteins. We identify highly conserved protein docking motifs within the N-terminal IDR of PEAK proteins and structurally characterize the phospho-dependent interaction of PEAK1/PEAK3 with the adapter protein Grb2. We provide molecular details of PEAK interactions with CrkII and identify a role for the scaffold protein 14-3-3 as a regulator of PEAK3 signaling. Lastly, we demonstrate that phosphorylation at PEAK3 S69 generates a high affinity 14-3-3 binding site. Binding of 14-3-3 to PEAK3 pS69 generates a highly stable PEAK3:14-3-3 dimer:dimer resulting in markedly reduced binding of CrkII to PEAK3 dimers. These findings contextualize why dimerization of PEAKs has a crucial function in signal transduction and demonstrate how signal specificity amongst the family is achieved. This exemplifies a 14-3-3-mediated molecular switch mechanism involving PEAK3/CrkII/Grb2, providing additional insights into the dynamic role of PEAKs in cell migration and invasion.

## Results

### The PEAK3 N-terminal IDR contains conserved interaction motifs

The N-terminal IDRs of PEAK family proteins contain numerous predicted SLIMs; several have confirmed roles in PEAK1/2 signaling[4,9,14,21,23,30,31], but it is likely that other important interaction sites remain to be validated, in particular those critical to PEAK3 signaling for which less is known. To further study the functionally relevant interactors of PEAK3, we first conducted an extensive bioinformatic analysis of the PEAK3 N-terminal IDR, which is comparatively shorter than those of PEAK1/2. We compared PEAK3 orthologs and direct conservation of this region with PEAK1 or PEAK2 (Fig. 1). This analysis highlighted two regions of interest. The first was a tyrosine/SH2 motif (YSNL; PEAK3[Y24]/PEAK1[Y1107]) that is highly conserved in PEAK3 and PEAK1 vertebrate orthologs (and absent in PEAK2 vertebrate orthologs) that matches the known phosphotyrosine consensus motif for the Grb2 SH2 domain (Grb2[SH2]; consensus motif: pYxNx) and the CrkII SH2 domain (CrkII[SH2]; consensus motif: pYxx(P/L), proline typically preferred) (Fig. 1a–d and Supplementary Fig. 1)[14,32]. The second region we identified was a "tandem site" that comprises two directly adjacent motifs that were both conserved: a PRM known to bind CrkII[NSH3] (consensus motif: PxLPxK)[4,14]; and a putative binding site for 14-3-3 proteins (mode 1 consensus motif: Rxx(pS/T)xP) (Fig. 1a–d). This tandem site was found to be conserved not only in

vertebrate orthologs of PEAK3, but also in vertebrate orthologs across the PEAK family (Fig. 1c and Supplementary Fig. 1). While cellular studies have shown that PEAK3 interactions with Grb2 and CrkII[4,14] are functionally important, neither have been directly characterized structurally or biophysically, nor have there been reports of a functional role for 14-3-3 interaction with PEAK proteins.

### The PEAK3[Y24]/PEAK1[Y1107] SH2 motif is phosphorylated by Src

We first focused on the molecular characterization of the conserved PEAK3[Y24]/PEAK1[Y1107] SH2 motif with Grb2. We recently confirmed phosphorylation of the PEAK3 TYSNL (pY24) motif in cells and showed this is required for recruitment of Grb2, and possibly CrkII and ASAP1[14], an Arf GTPase-activating protein that regulates cytoskeletal remodeling and is associated with tumor progression and invasiveness[31,33,34]. In MCF-10A cells, phosphorylation of PEAK3 at this site is blocked by treatment with either the SFK inhibitor eCF506 or the SFK/Abl inhibitor dasatinib, suggesting that this site is dependent on SFK activity[14]. We recently also described the expression and purification of full length recombinant PEAK3 (PEAK3[FL]) and N-terminally truncated forms of PEAK1 (PEAK1[IDR1] that includes PEAK1[Y1107] site, residues 1082–1746) and PEAK2 (PEAK2[IDR1], residues 802–1406) from insect cells (see "Methods")[35]—all of which also include the tandem motif in the IDR in addition to SHED/PsK domains. Building on this, we performed an in vitro kinase assay with purified recombinant PEAK3[FL] and PEAK1[IDR1] proteins followed by tandem mass spectrometry (MS/MS) analysis of tryptic peptides. We found that Src, but not Abl, can phosphorylate this conserved SH2 motif on PEAK3[FL] (pY24) and PEAK1[IDR1] (pY1107) corroborating our recently reported cellular data[14] (Fig. 2a).

### Phosphorylated PEAK3[Y24]/PEAK1[Y1107] SH2 motif binds Grb2[SH2]

We next sought to obtain structural data to better understand the mode of interaction of PEAK3 with Grb2. To do this we utilized a synthetic 7-mer pY phospho-peptide encompassing this SH2 motif (T**pY**SNLGQ; corresponding to PEAK3[23–29]-pY24 and PEAK1[1106–1112]-pY1107), hereafter "SH2-pY peptide". We have previously shown by isothermal titration calorimetry (ITC) that this SH2-pY peptide binds to both full length Grb2 (Grb2[FL], $K_D$ 2.6 μM) and CrkII (CrkII[FL], $K_D$ 7.8 μM)[14].

Using an approach developed for apo-Grb2[FL] [36], we succeeded in crystallizing a complex of human Grb2[FL]:PEAK SH2-pY peptide and solved an X-ray structure to a resolution of 2.7 Å (Table 1, Fig. 2b and Supplementary Fig. 2a–c). In our structure, Grb2[FL] is present as a dimer as seen in the structure of apo-Grb2[FL] (Protein Data Bank (PDB) ID: 1GRI)[36], with the SH2-pY peptide bound to the SH2 domain of only one copy of Grb2 (chain B) (Fig. 2b and Supplementary Fig. 2a–c). Close crystal packing within the asymmetric unit leaves only this SH2 site (chain B) available for phosphopeptide binding (Supplementary Fig. 2a–c). Our structure confirms that the PEAK3/PEAK1 SH2-pY peptide adopts a characteristic β-turn, with the expected recognition of the phosphotyrosine-moiety and the specificity-determining (+2) N residue forming three important hydrogen bonds to the Grb2 backbone (R86, H107 and L109), consistent with other Grb2[SH2]:pYxNx peptide structures (e.g., PDB ID: 1TZE)[37]. Whilst no density was apparent for the C-terminal (-GQ) residues of the peptide, we noticed a potential surface groove directly adjacent on Grb2[SH2] (Fig. 2b). In our orthologue multiple sequence alignment (MSA) data there is an extended region of high sequence conservation in PEAK3 and PEAK1 C-terminal to this SH2 motif (Fig. 2c, d). This corresponded to a predicted helical structure in AlphaFold2 (AF2)[38,39] models of PEAK3 and PEAK1 (Fig. 2c, d). We were thus interested to evaluate whether this extended region could confer additional CrkII[SH2]/Grb2[SH2] binding affinity or selectivity. SPR studies were conducted with the shorter consensus pY SH2 motif phosphopeptide, or longer PEAK3[23–38]pY24 and PEAK1[1106–1121]pY1107 phosphopeptides to measure binding to

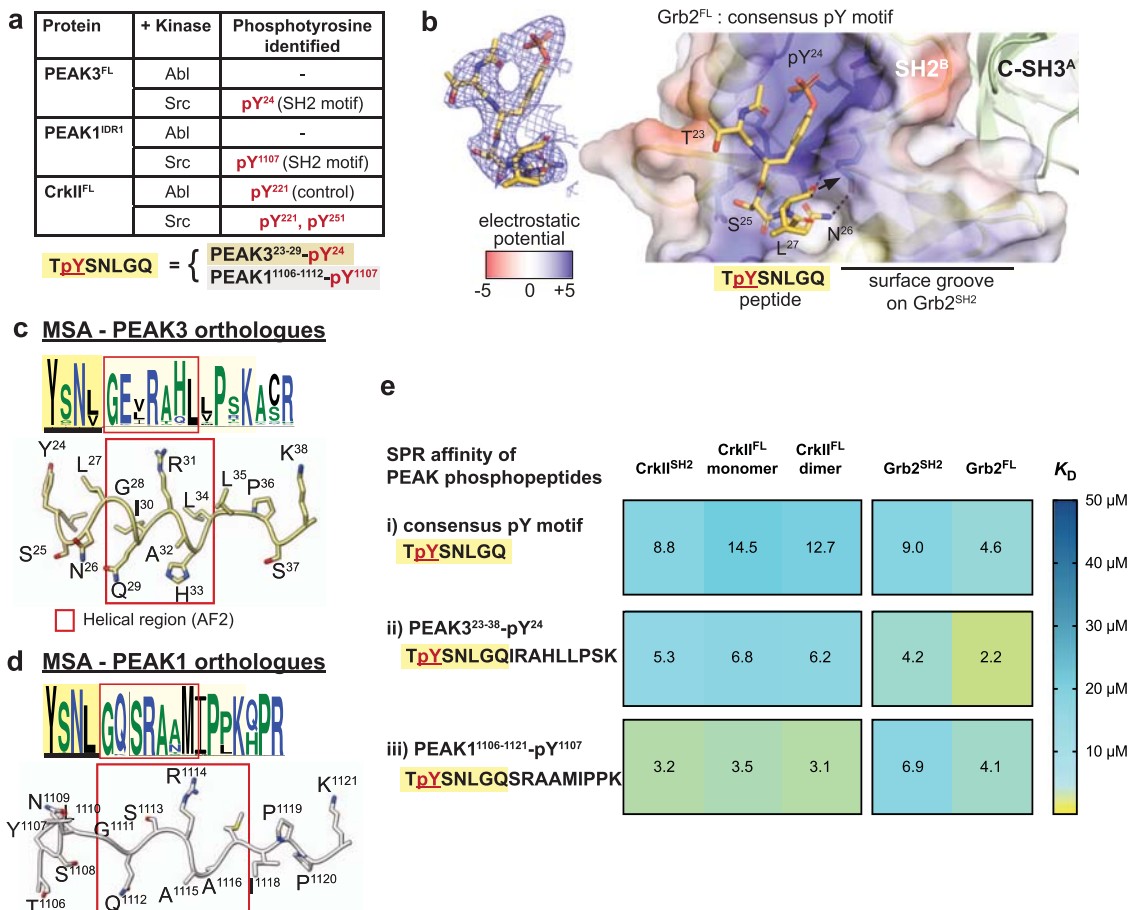

**Fig. 2 | Structural and biophysical analysis of PEAK/Grb2$^{SH2}$ interaction.**
**a** TYSNL site of PEAK3$^{FL}$ (Y221) and PEAK1$^{IDR1}$ (Y1107) can be phosphorylated by Src (MS/MS analysis of tryptic peptides following in vitro kinase assay of PEAK3$^{FL}$ and PEAK1$^{IDR1}$ are available in source data). Both Abl and Src can phosphorylate CrkII$^{Y221}$, a known Abl phosphorylation site[84], demonstrating that both Abl and Src kinases are active. **b** Zoom in structure of the PEAK SH2-pY peptide bound to the Grb2$^{FL}$ dimer, with the PEAK phosphopeptide shown in yellow and Grb2$^{FL}$ Chain B (peptide bound chain) shown in surface representation, colored by electrostatic surface potential calculated using the APBS Electrostatics plugin[85] for PyMOL (Schroedinger) (blue = positive, white = hydrophobic, red = negative). Inset shows a 2Fo·Fc electron density map (blue mesh, contoured at 1.0σ) showing the final modeled PEAK3/PEAK1 consensus phosphopeptide (TpYSNLGQ, yellow sticks, underlined

residues modeled). See also Supplementary Fig. 2. **c, d** PEAK3$^{23–38}$ and PEAK1$^{1106–1121}$ regions alongside full MSA/Web Logo of orthologs for the corresponding sequence, showing a section of high sequence conservation (pale yellow box) adjacent to the TYSNL SH2 site (yellow box, underlined) including a region predicted in AF2 to adopt helical secondary structure (red box). **e** Tabulated SPR-determined mean steady-state binding affinity values ($K_D$) for PEAK phosphotyrosine peptides (consensus pY-SH2 motif and extended peptides PEAK3$^{23-38}$pY24 and PEAK1$^{1106–1121}$-pY1107) binding to CrkII$^{SH2}$, full-length CrkII (CrkII$^{FL}$) monomer or dimer, Grb2$^{SH2}$ or full length Grb2 (Grb2$^{FL}$). Values are colored as a heat map to illustrate trends in affinity. Data represent mean values from n = 3 independent titrations. See Supplementary Fig. 2d for representative SPR sensorgrams and Supplementary Data 1 for full SPR tabulated data and sensorgrams.

immobilized full length Grb2 and CrkII (monomer and dimer) or individual SH2 domains. Together, these data indicate that these extended regions of PEAK3 and PEAK1 confer additional affinity and selectivity to the interactions, with PEAK3$^{23-38}$pY24 exhibiting tighter binding affinity and selectivity toward Grb2$^{SH2}$/Grb2$^{FL}$, whereas PEAK1$^{1106-1121}$pY1107 the additional region enables tighter binding to CrkII$^{SH2}$/CrkII$^{FL}$, but does not substantially alter binding to Grb2 (Fig. 2e and Supplementary Fig. 2d).

## PEAK proteins bind CrkII$^{NSH3}$ with similar affinity
Having made significant inroads to biophysically and structurally elucidate Grb2/CrkII binding at the PEAK3 SH2 site, we next turned our attention to the conserved "tandem site" we identified on PEAKs, containing a CrkII$^{NSH3}$ PRM and putative 14-3-3 binding site. Cellular studies show qualitatively that all PEAKs can interact with CrkII, requiring: (1) the conserved PRM within the "tandem site" (matching the consensus for CrkII$^{NSH3}$; PxLPxK); as well as (2) SHED-mediated PEAK dimerization[4,14,30]. However, PEAK family members have non-identical sequences at this tandem site and to our knowledge no study

has yet assessed the relative affinity of each site for CrkII$^{NSH3}$, which may assist to contextualize the hierarchy of these interactions.

We therefore addressed this by generating a series of synthetic peptides that encompass the PRM from the tandem site of each PEAK protein—PEAK3 (residues 54–66), PEAK1 (residues 1150–1162), PEAK2 (residues 809–821)—as well as another PRM in PEAK2 similar to the consensus sequence, PEAK2 (residues 709-721), and determined their affinity toward the immobilized CrkII$^{NSH3}$ domain by Surface Plasmon Resonance (SPR) (Fig. 3a, Supplementary Fig. 3a and Supplementary Data 2). We confirm that PEAK1$^{1150–1162}$, PEAK2$^{809–821}$ and PEAK3$^{54–66}$ peptides all bind CrkII$^{NSH3}$ with comparable affinity and fast on/off kinetics (measured dissociation constant, $K_D$ 0.9–2.4 μM). Whilst the affinity of PRM/SH3 interactions can vary (1–100 μM range)[40], this is comparable to CrkII$^{NSH3}$ affinity for other PRMs that also closely match the CrkII$^{NSH3}$ consensus sequence, such as the Abl$^{758}$ PRM (reported $K_D$ 1.7 μM)[41]. We also show that PEAK2$^{709-721}$ represents a potential second, lower affinity binding site for CrkII$^{NSH3}$ on PEAK2 ($K_D$: 13 μM) (Fig. 3a and Supplementary Data 2). Comparison of the PEAK3$^{54–66}$ peptide sequence to the crystal structure of the

**Table 1 | Data collection and refinement statistics**

| | Grb2$^{FL}$:PEAK SH2-pY peptide (PDB 8DGO) | 14-3-3ε:PEAK3$^{tandem}$-pS69 (PDB 8DGP) | 14-3-3ε:PEAK1$^{tandem}$-pT1165 (PDB 8DGM) | 14-3-3ε: PEAK2$^{tandem}$-pS826 (PDB 8DGN) |
|---|---|---|---|---|
| **Data collection**[a] | | | | |
| Space group | $P4_3$ | $P4_1$ | $P6_222$ | $P6_222$ |
| Cell dimensions | | | | |
| $a, b, c$ (Å) | 89.31, 89.31, 94.83 | 155.53, 155.53, 58.06 | 91.42, 91.42, 139.87 | 92.13, 92.13, 137.44 |
| a, b, g (°) | 90, 90, 90 | 90, 90, 90 | 90, 90, 120 | 90, 90, 120 |
| Resolution (Å) | 41.88–2.30 (2.38–2.30)[b] | 49.19–2.70 (2.80–2.70) | 45.72–3.20 (3.32–3.20) | 46.07–3.16 (3.27–3.16) |
| $R_{merge}$ | 0.079 (1.617) | 0.256 (1.885) | 0.104 (1.601) | 0.128 (1.673) |
| $I/\sigma I$ | 12.81 (1.11) | 9.74 (1.43) | 22.22 (1.93) | 18.79 (2.02) |
| Completeness (%) | 99.3 (93.9) | 99.6 (96.1) | 99.84 (99.83) | 99.75 (98.37) |
| Redundancy | 6.9 (6.4) | 14.0 (13.5) | 18.8 (19.3) | 18.9 (19.0) |
| **Refinement** | | | | |
| Resolution (Å) | 41.88–2.30 | 49.18–2.70 | 45.72–3.20 | 46.07–3.16 |
| No. reflections | 32,851 | 38,504 | 6142 | 6349 |
| $R_{work}/R_{free}$ | 0.1789/0.2232 | 0.1873/0.2330 | 0.2066/0.2661 | 0.2419/0.2947 |
| No. atoms | 3679 | 7663 | 1499 | 1510 |
| Protein | 3569 | 7512 | 1491 | 1510 |
| Ligand/ion | 0 | 35 | 8 | 0 |
| Water | 110 | 128 | 0 | 0 |
| $B$-factors | 73.44 | 60.89 | 102.97 | 106.78 |
| Protein | 73.91 | 61.03 | 103.00 | 106.78 |
| Ligand/ion | – | 89.86 | 98.29 | – |
| Water | 58.26 | 47.85 | – | – |
| R.m.s. deviations | | | | |
| Bond lengths (Å) | 0.008 | 0.006 | 0.009 | 0.003 |
| Bond angles (°) | 0.99 | 0.77 | 1.17 | 0.53 |
| $B$-factors by Chain (residues modeled) | **Grb2$^{FL}$:**<br>A: 67.74 (1–217)<br>B: 79.44 (1–217)<br>**PEAK3/1 SH2-pY:**<br>C: 97.74 (23–27)<br>(Residue numbering for human PEAK3) | **14-3-3ε:**<br>A: 41.58 (1–233)<br>B: 55.79 (2–234)<br>C: 71.25 (1–234)<br>D: 77.81 (1–234)<br>**PEAK3$^{tandem}$-pS69:**<br>E: 62.76 (66–72)<br>F: 59.53 (65–73)<br>G: 75.67 (65–72)<br>H: 73.70 (65–72) | **14-3-3ε:**<br>A: 104.67 (33–232)<br>**PEAK3$^{tandem}$-pS69:**<br>E: 118.22 (66–72) | **14-3-3ε:**<br>A: 107.95 (33–232)<br>**PEAK3$^{tandem}$-pS69:**<br>E: 113.87 (66–72) |

[a]Data are from one crystal for each structure.
[b]Values in parentheses are for the highest-resolution shell.

Abl PRM$^{758}$ with CrkII$^{NSH3}$ (PDB: 5IH2)[41] (Fig. 3b) highlights the role that the PEAK3 Lys62 residue (−3 position) and the complementary electrostatic potential play in determining CrkII$^{NSH3}$ specificity, as previously described for Abl[41,42].

**Avidity enables stable PEAK interactions with CrkII**
We next looked to examine the role of dimerization in CrkII binding to PEAKs. As our initial SPR studies had utilized isolated peptides and only the CrkII$^{NSH3}$ domain, we shifted to utilize recombinant CrkII$^{FL}$ and dimeric PEAK1, PEAK2 and PEAK3. CrkII$^{FL}$ was expressed and purified from *E. coli* as described previously[14]. Interestingly, whilst the majority of CrkII$^{FL}$ is monomeric, -10% of the total isolated protein purifies as a stable dimer (Supplementary Fig. 3b, c). Dimerization has been described for other adapters such as Grb2[43] and the related protein CrkL (mediated by the C-terminal SH3 domain[44]), however to our knowledge this has not been reported to date for CrkII. We confirm that, in contrast to CrkL, the dimerization of CrkII$^{FL}$ we observe is mediated by the SH2 domain, as recombinant C-terminally truncated constructs CrkII$^{\Delta CSH3}$ (SH2-NSH3) and CrkII$^{SH2}$ each form a similar proportion of dimer (Supplementary Fig. 3b, c). Further stability test conducted on purified CrkII$^{FL}$ monomers and dimer indicates that both

exist as stable states (Supplementary Fig. 3c). As MS analysis of both monomeric and dimeric CrkII$^{FL}$ shows no PTMs (see Source Data) we hypothesize the dimer is formed by a domain-swap within the SH2 domain.

We next focused on PEAK1 and PEAK2, as the SHED-PsK domains of these have previously been purified and crystallized[18,35] and utilized longer constructs, PEAK1$^{IDR1}$ and PEAK2$^{IDR1}$, harboring the "tandem motif"[35]. We recently reported the expression/purification of these longer constructs from insect cells[35]. Consistent with our previous findings for PEAK1/2[18], PEAK1$^{IDR1}$ and PEAK2$^{IDR1}$ purify as a homodimers (Fig. 3c, d and Supplementary Fig. 3d). Purified PEAK1$^{IDR1}$ and PEAK2$^{IDR1}$ were then analyzed for their ability to interact with either monomeric or dimeric CrkII$^{FL}$ on SEC followed by SDS-PAGE (Fig. 3c, d and Supplementary Fig. 2). Strikingly, whilst the PEAK2$^{IDR1}$ dimer:CrkII$^{FL}$ monomer complex eluted as two peaks at the expected elution volume for each component, consistent with the anticipated low-affinity complex (micromolar range with fast on/off kinetics), the PEAK2$^{IDR1}$ dimer:CrkII$^{FL}$ dimer complex eluted earlier, indicative of a stable interaction (Fig. 3c). SDS-PAGE confirmed that this complex contained both PEAK2$^{IDR1}$ and CrkII$^{FL}$ dimers (Fig. 3c). Similar results were obtained for PEAK1$^{IDR1}$ dimer:CrkII$^{FL}$

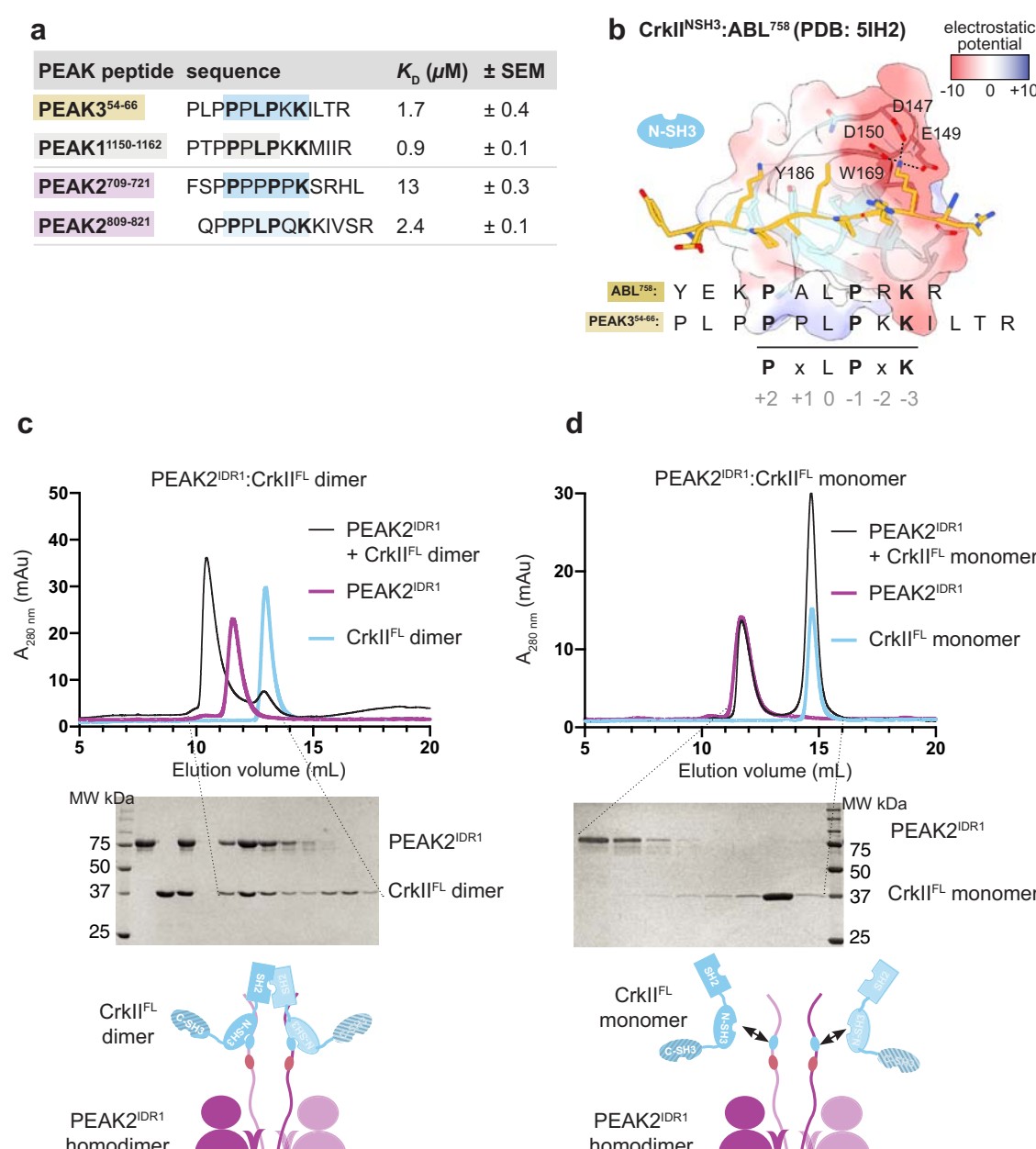

**Fig. 3 | Structural and biophysical analysis of PEAK/CrkII$^{NSH3}$ interaction. a** SPR data summarizing measured steady-state binding affinity ($K_D$) of synthetic PEAK PRM peptides toward immobilized CrkII$^{NSH3}$ domain. **b** Depiction of the published CrkII$^{NSH3}$:Abl$^{758}$ structure (PDB: 5IH2)[41], showing key residues within the CrkII$^{NSH3}$ consensus motif in Abl$^{758}$ (peptide shown in gold stick representation) crucial for high affinity binding to CrkII$^{NSH3}$ (main chain light blue; CrkII$^{NSH3}$ surface is colored by electrostatic surface potential calculated using UCSF Chimera v 1.16; blue = positive, white = hydrophobic, red = negative). Aligned is the sequence of PEAK3$^{54-66}$, showing key conserved residues of the CrkII$^{NSH3}$ motif. **c, d** Interaction studies with recombinant PEAK1$^{IDR1}$/PEAK2$^{IDR1}$ and CrkII$^{FL}$ underscore the role of avidity for high affinity binding of CrkII$^{FL}$ to PEAK dimers; **c** Incubation of PEAK2$^{IDR1}$ with CrkII$^{FL}$ dimer at a 1:1.3 ratio results in a complex formation while (**d**) incubation with CrkII$^{FL}$ monomer does not result in a complex, as confirmed by SDS-PAGE analysis of SEC eluted fractions ($n = 3$, independent experiments). Plots of the CrkII$^{FL}$ dimer and monomer are shown in light blue; PEAK2$^{IDR1}$ in pink; and the complex of PEAK2$^{IDR1}$ dimer:CrkII$^{FL}$ dimer in black (see Supplementary Fig. 3d for PEAK1$^{IDR1}$ dimer:CrkII$^{FL}$ dimer complex).

dimer (Supplementary Fig. 3d). SEC coupled to Multi-Angle Light Scattering (SEC-MALS) analysis of the purified PEAK2$^{IDR1}$ dimer:CrkII$^{FL}$ dimer complex estimated a mass of 192 kDa, close to the expected mass of 198 kDa for a stoichiometric (2:2) PEAK2$^{IDR1}$ dimer:CrkII$^{FL}$ dimer complex (Supplementary Fig. 3e). The marked differences in complex stability observed between binding of dimeric and monomeric CrkII$^{FL}$ to dimeric PEAK1 and PEAK2 reveal an important role for avidity, helping to rationalize the dimerization requirement observed in cellular studies[45].

## PEAK3 binds 14-3-3 via the tandem site 14-3-3 motif (pS69)

We next wanted to extend these studies to recombinant PEAK3$^{FL}$. Interestingly, PEAK3$^{FL}$ from insect cells was purified as a stable heteromeric complex with two additional interactors, identified by MS as insect cell derived 14-3-3 proteins (14-3-3ε,ζ heterodimer) (Fig. 4a)[35]. This was consistent with our SLIM identification of a highly conserved putative 14-3-3 motif present in the PEAK3 IDR. Co-purification of 14-3-3 from insect cells has been observed for other kinases with 14-3-3 binding sites (e.g., BRAF, LRKK2)[27,46–48]. Proteomic analysis of the

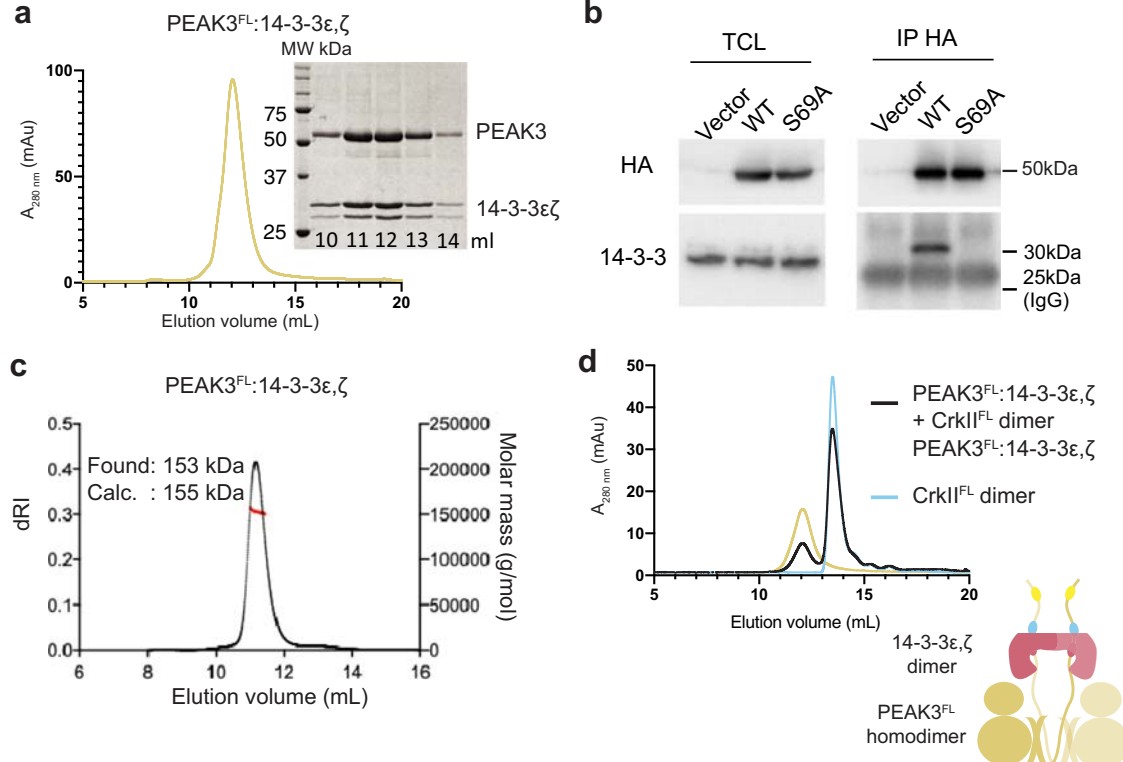

**Fig. 4 | PEAK3:14-3-3 forms a stable high affinity heterocomplex. a** Recombinant PEAK3[FL] from insect cells elutes on size exclusion chromatography (SEC; S200 10/300) as a high affinity stoichiometric complex of dimeric PEAK3 with a 14-3-3ε,ζ heterodimer, results supported by SDS-PAGE analysis of eluted fractions ($n = 3$, independent experiments) (see also Supplementary Fig. 4a, source data). **b** Immunoprecipitation of PEAK3 WT and PEAK3 S69A showing S69 is required for 14-3-3 co-immunoprecipitation from cells ($n = 3$ biologically independent samples, see Supplementary Fig. 4a for biological repeats). **c** SEC-MALS analysis of PEAK3[FL]–14-3-3ε,ζ heterodimer complex confirming the experimentally determined mass closely matches the mass of 155 kDa expected for a stoichiometric dimer:dimer complex ($n = 3$, independent experiments). **d** SEC profile (S200 10/300) of recombinant PEAK3[FL]–14-3-3ε,ζ heterodimer pre-incubated with CrkII[FL] dimer in a 1:1.2 molar ratio showing no complex formation ($n = 3$, independent experiments). Source data are provided as a source data file.

purified PEAK3[FL]:14-3-3ε,ζ complex confirmed that recombinant PEAK3 purified from insect cells was phosphorylated only at the 14-3-3 motif (pS69) within the tandem site (Supplementary Fig. 4a). Despite apparent conservation of this tandem 14-3-3 motif across the PEAK family, we observed differences in 14-3-3 interaction and phosphorylation of this motif between PEAK family members. Proteomic analysis of insect cell derived PEAK2[IDR1] showed phosphorylation of the tandem site 14-3-3 motif (RAASSP), whereas phosphorylation at this motif was not observed for PEAK1[IDR1] (RANTEP) (Supplementary Fig. 4a). Co-purification with 14-3-3 was only observed for PEAK3[FL] (Supplementary Fig. 4a). To complement these data and further characterize the contribution of PEAK3 pS69 in recruiting 14-3-3 in cells, we performed anti-HA IPs from HEK293 cells expressing HA-tagged PEAK3[FL] WT (HA/PEAK3[FL]) and PEAK3[FL]-S69A mutant (HA/PEAK3[FL]-S69A; Fig. 4b and Supplementary Fig. 4b). These experiments confirmed that while HA-tagged PEAK3[FL] WT co-immunoprecipitates with endogenous 14-3-3, this interaction was abolished with HA-tagged PEAK3[FL]-S69A, indicating that S69 is the primary cellular 14-3-3 interaction site for PEAK3.

### PEAK3:14-3-3 form a stable dimer:dimer heterocomplex
We recently showed that PEAK3 can form homodimers in a cellular context[14]. To study the effect of PEAK3 dimerization on 14-3-3 binding in cells, we co-expressed in HEK293 cells Flag-tagged PEAK3[FL] WT with HA-tagged versions of PEAK3[FL] WT or PEAK3[FL] dimerization mutants (L146E, A436E, C453E)[14,18]. Immunoprecipitation and western blotting experiments confirmed that the inability of these mutants to undergo homodimerization prevents recruitment of 14-3-3 (Supplementary

Fig. 4c), demonstrating that 14-3-3 association to PEAK3 is also dependent on PEAK3 dimerization. These findings build on our previous data[14] (and reports from the Jura laboratory[4]) demonstrating the dimerization dependent recruitment of other interactors such as CrkII and ASAP1 to PEAK3.

We next used SEC-MALS to confirm the stoichiometry of our recombinant PEAK3[FL]:14-3-3ε,ζ heteromeric complex, as homo- or heterodimers of 14-3-3 will contain two separate phosphopeptide binding sites. A single symmetrical peak with an experimental mass of 153 kDa was observed, suggesting a stable PEAK3[FL]:14-3-3ε,ζ dimer:dimer complex (expected mass 155 kDa) (Fig. 4c). We next performed SEC interaction studies of the PEAK3[FL]:14-3-3ε,ζ complex with the CrkII[FL] dimer. Interestingly, while CrkII[FL] complexes with PEAK1[IDR1] or PEAK2[IDR1] (Fig. 3c, d), the PEAK3[FL]:14-3-3ε,ζ complex showed no complex formation with CrkII[FL], despite the presence of the CrkII[NSH3] binding motif in the PEAK3[FL] construct (Fig. 4d).

### PEAK3 binds 14-3-3 at the tandem site in a phosphodependent manner
Given the proximity of the 14-3-3 motif and CrkII[NSH3] motifs within the tandem site, we next wanted to compare the binding affinity of 14-3-3 and CrkII[NSH3] at each respective site (Fig. 5 and Supplementary Fig. 5). We generated peptides of PEAK3, PEAK1 or PEAK2 that encompass the tandem site (CrkII[NSH3] and 14-3-3 motif), phosphorylated or non-phosphorylated at the 14-3-3 motif and conducted biophysical interaction studies by SPR with purified 14-3-3 and CrkII proteins (Supplementary Fig. 5c and Supplementary Data 3 and 4). We selected, as an initial subset, four human 14-3-3 isoforms (γ, ε, η, σ) out of the seven

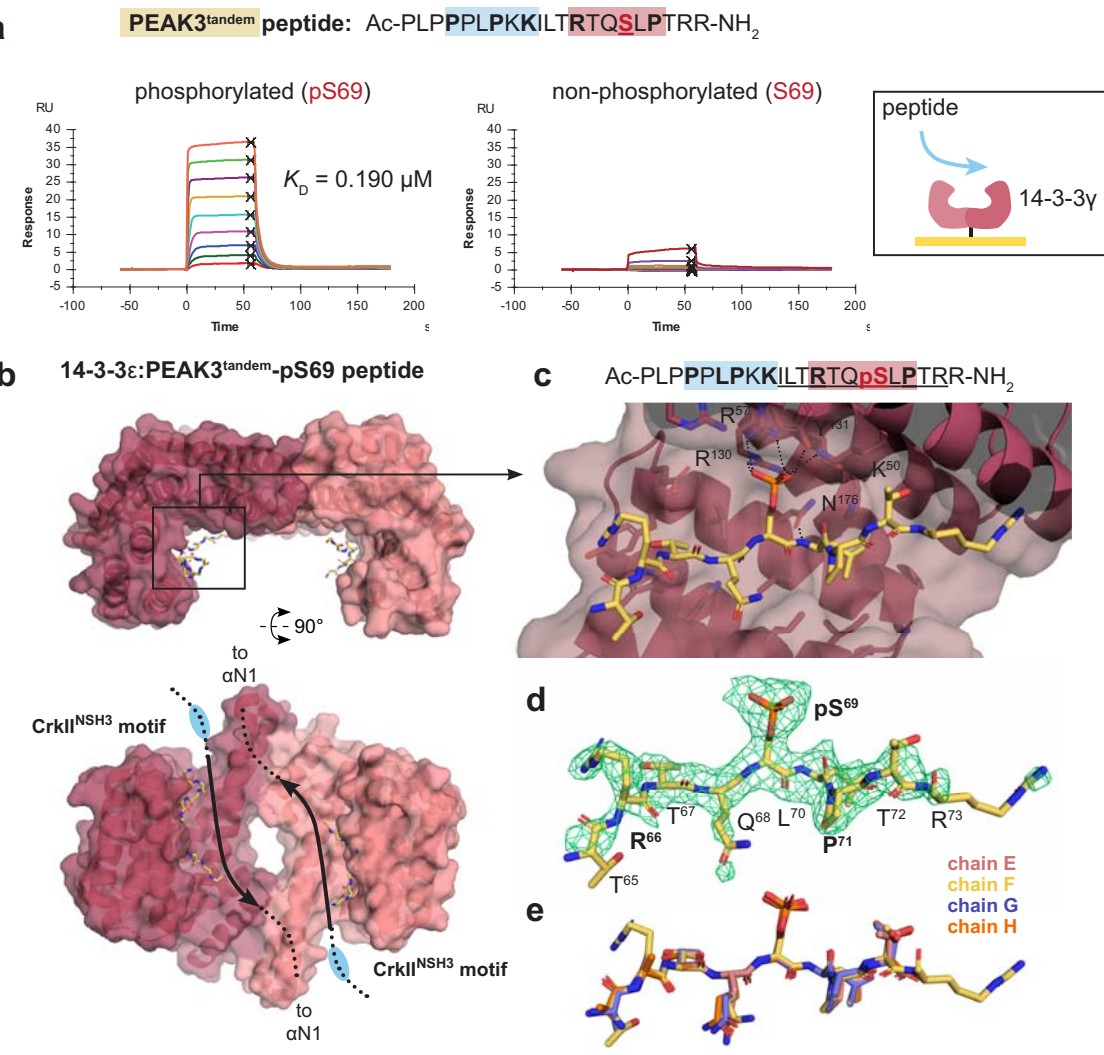

**Fig. 5 | Structural and biophysical analysis of PEAK3/14-3-3 interaction.**
**a** Binding of PEAK3$^{tandem}$-pS69 (left) and PEAK3$^{tandem}$-S69 peptides (right) to 14-3-3γ as measured by SPR. **b** Overall structure of 14-3-3ε: PEAK3$^{tandem}$-pS69 peptide showing 14-3-3 dimer antiparallel arrangement with each monomer (pink cartoon/surface) bound to a single copy of the PEAK3$^{tandem}$-pS69 peptide (yellow sticks). Chain A of 14-3-3ε is in dark pink and chain B is in light pink. **c** Zoom in highlighting peptide groove and showing the central pS69 residue of the PEAK3 phosphopeptide interacting with K50, R57, Y131 and R130. **d** Unbiased Fo-Fc omit map (green mesh, contoured at 3.0σ) showing peptide density prior to modeling and final modeled PEAK3$^{tandem}$-pS69 phosphopeptide (yellow sticks). **e** Superposition of the peptide modeled in each 14-3-3 monomer (chains A–D, sticks) in the asymmetric unit.

human isoforms, as those were most abundant in proteomic interaction analysis of PEAK proteins in cells[4,20]. SPR analysis of PEAK3 tandem peptides confirmed that phosphorylated PEAK3$^{tandem}$-pS69 peptide binds with high affinity to immobilized 14-3-3γ (SPR dissociation constant, $K_D$ = 0.19 μM), whereas the corresponding non-phosphorylated PEAK3$^{tandem}$-S69 peptide shows negligible binding ($K_D$ > 10 μM) (Fig. 5a).

We next compared the SPR binding of PEAK3, PEAK1 and PEAK2 tandem peptides to human 14-3-3γ, 14-3-3ε, 14-3-3η and 14-3-3σ. For each isoform, phosphodependent binding was observed and the PEAK3$^{tandem}$-pS69 phosphopeptide consistently showed ~10-fold tighter affinity ($K_D$ ~ 0.16–0.78 μM) than those of PEAK1$^{tandem}$-pT1165 or PEAK2$^{tandem}$-pS826 ($K_D$ ~ 1–10 μM) (Supplementary Fig. 5c). This difference in affinity was also apparent in the comparatively slower dissociation kinetics for 14-3-3:PEAK3$^{tandem}$-pS69 complexes relative to PEAK1 or PEAK2 tandem peptides (Supplementary Fig. 5c and Supplementary Data 4).

To gain molecular details of the PEAK3 interaction with 14-3-3, we determined the crystal structure of a 14-3-3ε:PEAK3$^{tandem}$-pS69 complex

to 2.5 Å resolution (Table 1). This complex crystallized with two 14-3-3ε dimers in the asymmetric unit (ASU), each bound to a single copy of the PEAK3$^{tandem}$-pS69 peptide. In each symmetric 14-3-3ε dimer, the PEAK3$^{tandem}$-pS69 peptide is present in the peptide binding groove in an overall antiparallel arrangement (Fig. 5b), with clear unbiased electron density for the central phosphoserine residue (PEAK3 pS69) recognized by 14-3-3ε as well as adjacent residues of the PEAK3 motif (~7 residues in total). Notably, interactions typical of such 14-3-3 complexes are observed: the PEAK3 pS69 phosphate group is coordinated by 14-3-3ε residues K50, R57, R130 and Y131 and 14-3-3ε D127 and N176 interact to enable N176 to form a hydrogen bond to the backbone amide (−NH) of PEAK3 L70. Each copy of the PEAK3$^{tandem}$-pS69 peptide within the ASU adopts a similar conformation (Fig. 5c–e, Supplementary Fig. 5b and Table 1). We also solved two crystal structures of 14-3-3ε: PEAK1$^{tandem}$-pT1165 and 14-3-3ε:PEAK2$^{tandem}$-pS826 complexes, which were lower resolution (3.1 Å) but similarly confirmed phosphorecognition and a canonical mode of binding. In each of the three 14-3-3:tandem peptide structures, no clear electron density was observed for the adjacent CrkII$^{NSH3}$ PRM likely due to flexibility and crystallographic averaging.

**a**

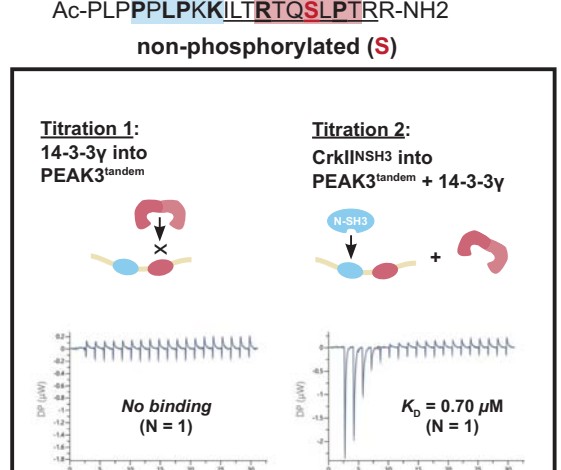

**b**

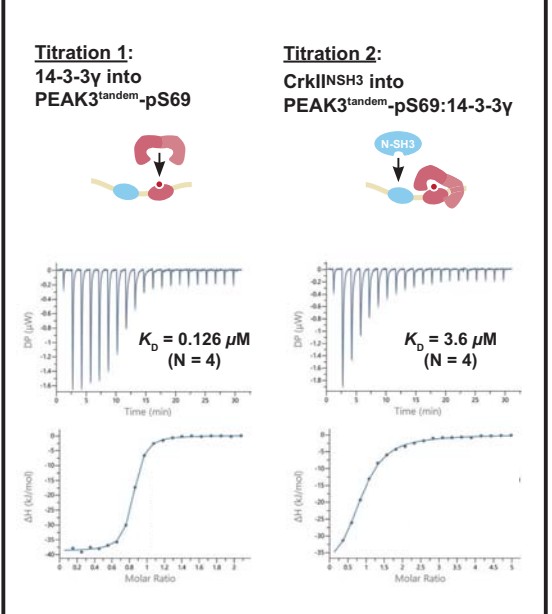

**Fig. 6 | 14-3-3γ binding to PEAK3<sup>tandem</sup> peptide reduces the affinity of CrkII<sup>NSH3</sup> binding to the adjacent CrkII motif. a** Sequential ITC binding studies using purified 14-3-3γ and purified CrkII<sup>NSH3</sup> domain and PEAK3<sup>tandem</sup>-S69 non-phosphorylated and **b** PEAK3<sup>tandem</sup>-pS69 phosphorylated peptides. **b** High affinity binding of 14-3-3γ to PEAK3<sup>tandem</sup>-pS69 peptide reduces the affinity of CrkII<sup>NSH3</sup> binding to the adjacent CrkII motif at the tandem site (negative cooperativity). See Supplementary Fig. 6a–d and Supplementary Data 5 for all ITC data.

### The 14-3-3 site on PEAK3 is a molecular switch for CrkII binding

To biophysically evaluate potential positive or negative cooperativity between these two adjacent binding sites in the PEAK3 tandem motif, we turned to sequential binding studies by ITC (Fig. 6a, b, Supplementary Fig. 6a–d and Supplementary Data 5). In these experiments, the PEAK3 tandem peptide (either phosphorylated or non-phosphorylated) was loaded in the cell and protein (either 14-3-3γ or CrkII<sup>NSH3</sup>) loaded in the syringe. First a titration of 14-3-3γ was measured, to saturate the first site on the peptide (binary experiment). The tandem peptide/14-3-3γ complex was retained in the cell and a second titration measured for CrkII<sup>NSH3</sup> in the presence of 14-3-3γ (ternary experiment) (Fig. 6). Corresponding sequential titrations were also conducted using buffer (first injection) followed by CrkII<sup>NSH3</sup> (second injection) (binary experiment). Direct SPR binding studies for each PEAK3<sup>tandem</sup> peptide to immobilized CrkII<sup>NSH3</sup> were also undertaken. The results for ITC and SPR binding experiments for the PEAK3<sup>tandem</sup> peptides are summarized in Table 2.

For the non-phosphorylated PEAK3<sup>tandem</sup> peptide, both SPR and ITC yield an affinity ($K_D$) of 0.60 μM for binding to CrkII<sup>NSH3</sup>, whilst no appreciable binding of this peptide to 14-3-3γ is observed. Similarly, the affinity of this peptide for CrkII<sup>NSH3</sup> was unaffected when pre-saturated with 14-3-3γ (ITC $K_D$ 0.70 μM). In contrast, phosphorylated PEAK3<sup>tandem</sup>-pS69 peptide showed strong binary binding to 14-3-3γ by either method (SPR $K_D$ 0.19 μM, ITC $K_D$ 0.13 μM), but modestly weaker binary binding to CrkII<sup>NSH3</sup> relative to the non-phosphorylated peptide (SPR $K_D$ 2.2 μM, ITC $K_D$ 1.1 μM). It is possible that the phosphate group reduces electrostatic complementarity to the adjacent negatively charged patch on the CrkII<sup>NSH3</sup> binding cleft[49]. When the phosphorylated PEAK3<sup>tandem</sup>-pS69 peptide was first pre-complexed with 14-3-3γ, binding of CrkII<sup>NSH3</sup> was further weakened (ITC $K_D$ 3.6 μM). This represents a 6-fold loss in affinity for the PEAK3 tandem peptide toward CrkII<sup>NSH3</sup> following pS69 phosphorylation and 14-3-3

binding, demonstrating negative cooperativity in these interactions at the tandem site. In the context of a dimeric scaffold (dimer:dimer of PEAK3:14-3-3ε,ζ) it is expected this negative cooperativity will be further pronounced due to avidity effects, which supports the inability of the CrkII<sup>FL</sup> dimer to form a complex with PEAK3:14-3-3ε,ζ via SEC.

Consistent with these biochemical and biophysical data, in our HEK293 cellular immunoprecipitation experiments using wildtype HA-tagged PEAK3<sup>FL</sup> and the PEAK3<sup>FL</sup>-S69A mutant, we observed that alongside the complete loss of 14-3-3 association for the PEAK3<sup>FL</sup>-S69A mutant there was a modest but significant increase in level of associated CrkII, relative to PEAK3<sup>FL</sup>-WT. This is also accompanied by a slight increase in Grb2 binding (Supplementary Fig. 4b). Importantly, this is despite the level of CrkII/Grb2associated with PEAK3<sup>FL</sup>-WT (or S69A) in these experiments likely being a composite of a number of multivalent interactions of these adapter proteins with multiple partners, including potentially interaction of CrkII with heterodimeric PEAK3/PEAK1 or PEAK3/PEAK2 via the CrkII<sup>NSH3</sup> site, or the PEAK3-pY24 motif via the CrkII<sup>SH2</sup>/Grb2<sup>SH2</sup> site, as we have shown previously[14]. To further demonstrate the impact of PEAK3/14-3-3/CrkII interaction in cells, we expressed PEAK3<sup>FL</sup>-WT and PEAK3<sup>FL</sup>-S69A in MCF-10A cells and assessed PEAK3<sup>FL</sup> tyrosine phosphorylation by immunoprecipitation and western blotting experiments using a pan pTyr antibody. We found that expression of PEAK3<sup>FL</sup>-S69A mutant results in a significant increase of PEAK3 pTyr phosphorylation compared to the WT protein (Fig. 7a). Our previous study demonstrated that mutation of Y24 abolished PEAK3 tyrosine phosphorylation and that the pY24/Grb2 interface represents a key PEAK3 signaling axis[14]. Taken together, these data indicate that abolishing 14-3-3 binding enhances CrkII binding and pY24/Grb2 mediated downstream signaling. As Grb2 signaling has a crucial role in regulating actin-based cell motility[25,50], we next directly assessed the impact of PEAK3 S69A mutation in this context.

**Table 2 | Summary of ITC sequential binding studies data**

| PEAK3 peptide, or peptide complex (peptide sequence) | ITC expt. type | CrkII^NSH3 | | | | 14-3-3γ | | | |
|---|---|---|---|---|---|---|---|---|---|
| | | SPR $K_D$ (μM) | N | ITC $K_D$ (μM) | N | SPR $K_D$ (μM) | N | ITC $K_D$ (μM) | N |
| **PEAK3**[54-66] peptide alone (PLP**PPL**P**KK**ILTR) | Binary | 1.7 ± 0.4 | 3 | ND | – | No binding | 3 | ND | – |
| **PEAK3**[tandem, 54-74] peptide alone (PLP**PPL**P**KK**ILT**R**TQ**SLP**TRR) | Binary | 0.60 ± 0.02 | 2 | 0.60 | 1 | No binding | | No binding | |
| **PEAK3**[tandem, 54-74]:14-3-3γ complex (PLP**PPL**P**KK**ILT**R**TQ**SLP**TRR) | Ternary | ND | | 0.70 | 1 | NA | | NA | |
| **PEAK3**[tandem, 54-74]pS69 peptide alone (PLP**PPL**P**KK**ILT**R**TQ**pSLP**TRR) | Binary | 2.2 ± 0.7 | 2 | 1.1 | 1 | 0.19 ± 0.04 | 4 | 0.130 ± 0.7 | 4 |
| **PEAK3**[tandem, 54-74]pS69:14-3-3γ complex (PLP**PPL**P**KK**ILT**R**TQ**pSLP**TRR) | Ternary | ND | | 3.6 ± 0.6 | 4 | NA | | NA | |

Refer to Supplementary Data 3–5 (SPR, ITC data). Values are mean, uncertainties are SD (N = 2) or S.E.M (N ≥ 3), from the listed number of independent experiments (N). Peptide residues from relevant consensus sequence motifs are bolded.

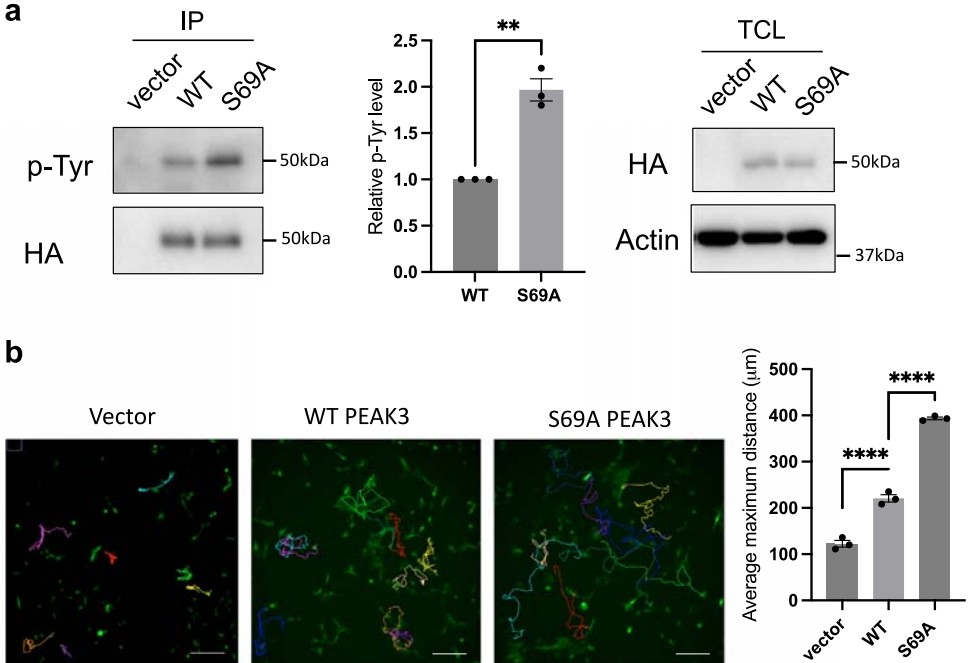

**Fig. 7 | S69A mutant increases PEAK3 p-Tyr and cell motility. a** S69A mutant increases PEAK3 p-Tyr. MCF-10A expressing WT and S69A PEAK3 cells were starved in −EGF medium overnight. Anti-HA IPs were prepared from denatured cell lysates and Western blotted as indicated. Data represent the mean ± S.E.M. of n = 3 independent experiments. **p < 0.01 (p = 0.0078), by ratio paired t-test, two tailed (Repeats shown in source data). **b** Effect of S69A PEAK3 on random cell motility in MCF-10A cells. Vector, HA-tagged WT or S69A PEAK3 were stably overexpressed in MCF-10A cells and random cell motility was determined by live cell tracking. Representative images are shown. Scale bar, 200 μm. The histogram indicates the average maximum displacement distance from the origin. ~40 cells were analyzed in each experiment. Data represent the mean ± S.E.M. of N = 3 independent experiments. ****p < 0.0001 by one-way ANOVA with Dunnett's multiple comparisons test (Vector vs. WT, p = 0.00090; WT vs. S69A, p = 0.000003), (Repeats shown in source data). Source data are provided as a Source Data File.

Strikingly, cell motility in PEAK3^FL-S69A expressing cells is significant increased, supporting the conclusion that 14-3-3 likely represents a key direct regulator of PEAK3 function and may inhibit PEAK3 pY24/Grb2 motility signaling (Fig. 7b).

**Modeling PEAK3/CrkII/Grb2 regulation via 14-3-3/CrkII tandem site**

To build a detailed model of PEAK/CrkII/Grb2 interactions, we used SPR binding studies using immobilized biotinylated full length or N-terminally truncated recombinant PEAKs (insect cell expressed human PEAK1^IDR1, PEAK2^IDR1, PEAK3^FL–14-3-3 complex and bacterially expressed PEAK3^FL) to measure the affinity of interactions with full-length interactors and relevant sub-domains (14-3-3γ, CrkII^FL monomer or dimer, Grb2^FL), both with or without prior phosphorylation of PEAKs by Src kinase (Fig. 8a and Supplementary Data 6).

These SPR studies confirmed the key interactions previously mapped using individual peptides. Notably, whilst insect cell expressed PEAK3^FL (S69 phosphorylated) was purified and immobilized as a complex with insect derived 14-3-3 (see "Methods"), it showed strong and reversible binding to injected recombinant human 14-3-3γ, confirming that endogenous insect 14-3-3 had dissociated from PEAK3^FL under the continuous flow conditions in SPR experiment. Consistent with PEAK phosphorylation state and measured affinity of individual tandem peptides (Supplementary Fig. 5c), only insect cell expressed human PEAK3^FL (and no other PEAKs tested, including human PEAK3^FL purified from *E.coli*) bound strongly to 14-3-3γ.

Additionally, phosphorylation of PEAKs by Src was required for Grb2^SH2 and CrkII^SH2 to bind, with Grb2^SH2 demonstrating tightest binding to PEAK3^FL, whilst CrkII^SH2 bound most tightly to PEAK1^IDR1 (Fig. 8b) consistent with the SPR results using individual pY

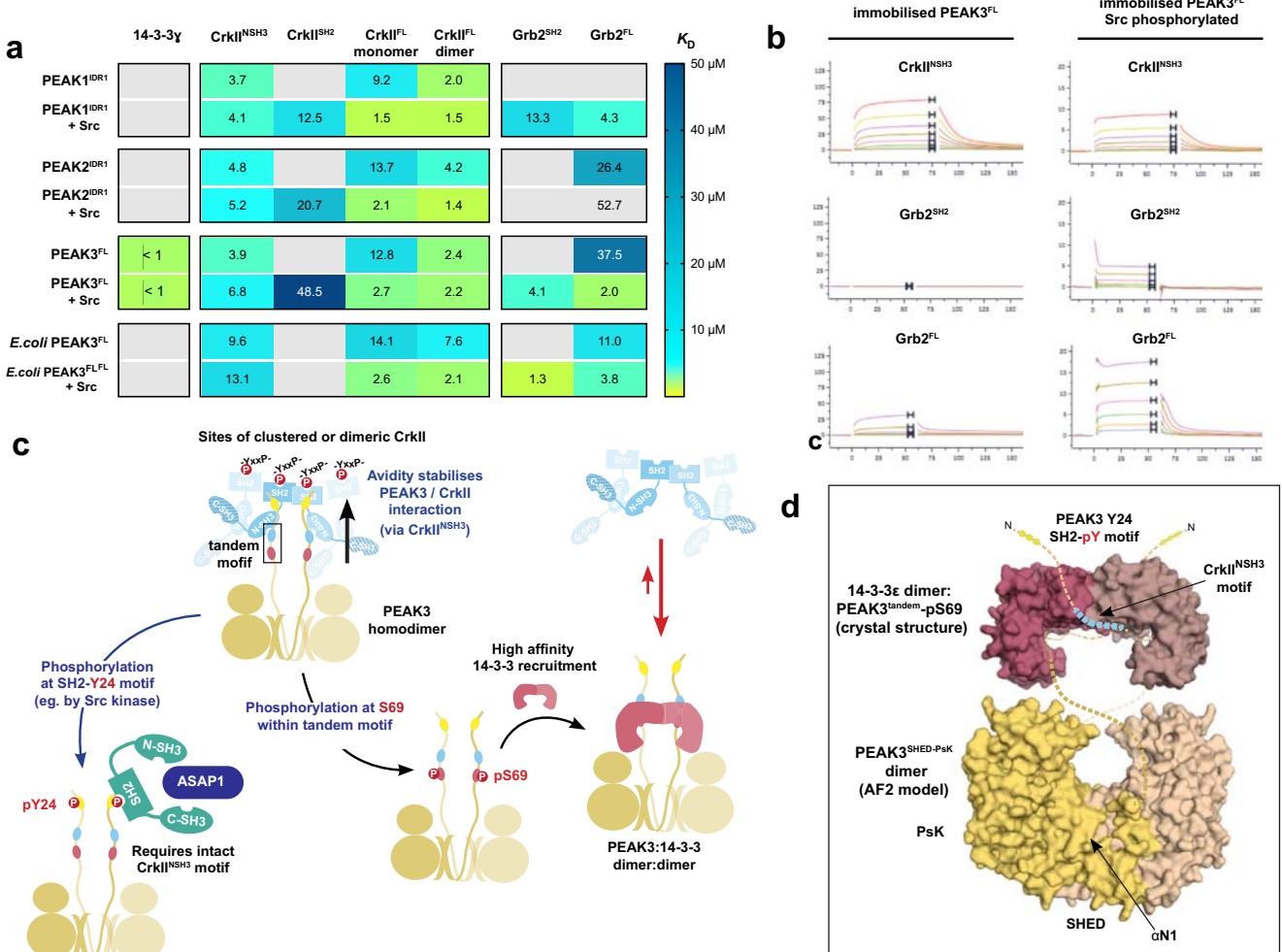

**Fig. 8 | Unified SPR analysis of PEAK protein interactions and proposed model for PEAK3/Grb2 and PEAK3/CrkII regulation via the tandem site 14-3-3 motif.**
**a** SPR binding contributions were measured for PEAK interactors (14-3-3γ, Grb2$^{FL}$ and CrkII$^{FL}$ and sub-domains) toward immobilized PEAK proteins with or without phosphorylation by Src. Tabulated mean SPR affinity values ($K_D$) are colored as a heat map to illustrate trends in affinity. Gray cells represent $K_D > 50\,\mu M$ or no detectable binding. **b** Representative SPR sensorgrams for key interactions of PEAK3$^{FL}$. **c** A unifying model of PEAK3 interactions. Avidity stabilizes PEAK3/CrkII interaction with clustered CrkII (interacting via the CrkII$^{NSH3}$ and tandem motif PRM of the PEAK3 dimer). CrkII is likely to be clustered by interaction with other adapters/scaffolds, such as SH2 domain-mediated binding to partner proteins containing multiple adjacent pYxxP motifs (e.g., present in known Src substrates p130cas/NEDD9, paxillin), but possibly also via dimerization (e.g., SH2-mediated as we identify for recombinant protein). Phosphorylation of PEAK3 at S69 creates a high-affinity binding site for 14-3-3, leading to the formation of a highly stable PEAK:14-3-3 dimer:dimer. Phosphorylation/binding of 14-3-3 to PEAK3 destabilizes

CrkII binding in the adjacent tandem site (negative cooperativity), disfavoring PEAK3/CrkII complexes. Possible outcomes are termination of PEAK3/CrkII signaling and/or alter PEAK3 sub-cellular localization. CrkII may facilitate colocalization of PEAK3 to regions with active Src kinase (e.g., p130cas at focal adhesions). Phosphorylation of PEAK3 at Y24 (e.g., by Src kinase) recruits Grb2 via its SH2 domain leading to downstream signaling that alters cell motility (e.g., via association with ASAP1 and PYK2 and potential PYK2/ASAP1-mediated activation of the PI3K/AKT pathway). Thus, PEAK3 S69A mutation abrogates 14-3-3 binding and promotes CrkII association and PEAK Y24 phosphorylation/Grb2 signaling, culminating in enhanced in cell motility. **d** Illustrative representation of a potential dimeric human PEAK3$^{FL}$:14-3-3ε complex to show relative scale and key sites of interaction. Depicted is the 14-3-3ε:PEAK3$^{tandem-pS69}$ crystal structure (PDB 8DGP) and a structural model of the PEAK3$^{SHED-PsK}$ dimer prepared from AlphaFold2 multimer modeling of a human PEAK3$^{FL}$ dimer[80,81]. Unmodelled IDR linker regions are depicted as dotted lines.

phosphopeptides from the SH2 motif (Fig. 2e). In line with studies using PEAK tandem site proline-rich motif peptides, the affinity of CrkII$^{NSH3}$ toward PEAK dimers was found to be relatively similar across all PEAKs ($K_D$ in low micromolar range) irrespective of Src phosphorylation state, reinforcing our understanding of the NSH3/PRM interaction as a primary PEAK-CrkII recruitment site (Fig. 8a).

The data also reveal contributions of multiple interaction domains to avidity in interaction with dimeric PEAK scaffolds. Notably, relative to its monomeric form, as observed in SEC studies, dimeric CrkII$^{FL}$ binds more tightly to all PEAKs (reflecting avidity of dimer:dimer interaction involving dual NSH3/PRM interfaces) (Fig. 8a). Additionally, whilst monomeric CrkII$^{FL}$ can already bind PEAKs (via NSH3), the

overall affinity of this interaction is markedly enhanced following phosphorylation of PEAKs by Src, reflecting the combined effect of simultaneous SH2 and NSH3 engagement. In contrast, for Grb2, significant binding to PEAKs is only observed following Src phosphorylation (and only for PEAK3$^{FL}$ and PEAK1$^{IDR1}$), confirming that direct Grb2 recruitment to PEAKs is primarily SH2-mediated and likely occurring via the consensus pY-SH2 motif we have identified and characterised.

Taken together, our data provide the molecular basis for a combined model of PEAK3 regulation, in which the "tandem site" we identified within the relatively short IDR of PEAK3 acts a molecular switch for CrkII binding via 14-3-3 (Fig. 8c). PEAK3, PEAK1 and PEAK2

(as homo or heterodimers) can all bind CrkII at the conserved tandem site ($K_D$ in low micromolar range with CrkII$^{NSH3}$ domain at each 1:1 interaction site on the PEAK dimer, but likely further enhanced by avidity where CrkII is clustered or dimeric) (Fig. 8a, b). Phosphorylation at PEAK3 S69 (tandem site) generates a high affinity binding site for dimeric 14-3-3 proteins ($K_D$ ~ 0.1–1 μM for at each site, further enhanced by avidity due to dimer/dimer interaction). This enables formation of a highly stable PEAK3:14-3-3 dimer:dimer complex (Fig. 8b, c). Due to negative cooperativity between 14-3-3 and CrkII$^{NSH3}$ binding at each tandem site, the PEAK3 interaction with CrkII is destabilized (Fig. 8c). 14-3-3 thus appears to be a key regulator of PEAK3 activity and modulates signaling via CrkII, with the crucial switching event being phosphorylation at the PEAK3 tandem site S69 motif. We propose that CrkII mediates recruitment of PEAK3 to sites of active Src/SFKs and thereby facilitates PEAK3 pY$^{24}$ phosphorylation and downstream signaling via Grb2 to enhance cell motility. This is consistent with observation that an intact PEAK3 binding site for CrkII$^{NSH3}$ is crucial for Src/SFK-mediated phosphorylation of PEAK3 and binding of Grb2 at the pY$^{24}$ site. Additionally, we show that overexpression of the PEAK3-S69A mutant that cannot bind 14-3-3 leads to enhanced levels of cellular pY$^{24}$ phosphorylation and cell motility. We therefore propose that 14-3-3 binding can act as a master regulator of PEAK3 signaling via both CrkII and Grb2 to downstream pathways (Fig. 8c).

## Discussion

Our studies illuminate several aspects of the dynamic regulation of PEAK scaffolding functions, particularly in the context of integrin- or EGFR-mediated signaling (Fig. 8 and Supplementary Fig. 7). We apply a combination of analytical techniques to characterize PEAK3 interactions with Grb2, CrkII and 14-3-3 via its N-terminal IDR, and the role of within the PEAK3 IDR. We demonstrate how dimerization-induced avidity effects and molecular crowding can each modulate SH3/PRM interactions, including how these mechanisms of interaction are conserved or differ across the PEAK family.

Firstly, we provide additional molecular insights to contextualize the interaction of PEAK3 and PEAK1 with Grb2/ASAP1[14,23]. We confirm that active Src, but not Abl, phosphorylates the conserved TYSNL SH2-motif (PEAK3$^{Y24}$, PEAK1$^{Y1107}$), supporting previous cellular studies[14], and thereby provides a binding site on either PEAK3 or PEAK1 for Grb2$^{SH2}$ or possibly CrkII$^{SH2}$ (Fig. 2). We provide a high-resolution structure of Grb2$^{FL}$ characterizing the binding of PEAK1/PEAK3 at this site and identify from AlphaFold2 models of PEAK1/PEAK3 an additional predicted helical region C-terminal to this consensus SH2-motif that appears to contribute additional Grb2$^{SH2}$/CrkII$^{SH2}$ binding affinity/selectivity (Fig. 2). This selectivity profile is also reflected for PEAK proteins binding to full-length Grb2 or CrkII (Fig. 8a). The observation that PEAK3 dimerization is required for Y$^{24}$ phosphorylation (and Grb2 association)[14] is also consistent with a model in which clustered CrkII assists to colocalize dimeric PEAK3 with an active kinase (e.g., Src/SFKs) for efficient Y$^{24}$ phosphorylation. Grb2 is also thought to undergo dynamic switching between dimeric (inactive) and monomeric (active, Sos-binding) states; a feature reported to be pivotal for Grb2 signaling in cancer[43]. Further studies are needed to examine whether PEAK dimers can impact this process also.

Secondly, we examine how NSH3/PRM interactions and avidity together mediate PEAK3 interaction with CrkII, a key player in FA turnover. Our work clearly places PEAKs as recruited scaffolds within this context at FAs (Fig. 7a, b), together with PYK2 and active Src, other important downstream mediators of integrin signaling[51,52]. We show that the conserved PRM in the PEAK IDR tandem site alone confers modest affinity for CrkII$^{NSH3}$ (SPR $K_D$ low micromolar range with a rapid off-rate; Figs. 3 and 8a), which is essentially equivalent for all PEAKs. However, we demonstrate using monomeric and dimeric forms of CrkII$^{FL}$ that SHED-dependent homodimerization (and by implication also hetero-dimerization of PEAKs)[4,17–20] enables enhanced stability of PEAK complexes with clustered/dimeric CrkII due to avidity (Figs. 3 and 8 and Supplementary Fig. 7b). These findings further support and rationalize data from this work and other reports[4,14] demonstrating that mutation of key residues of the PEAK3 dimer interface disrupts PEAK3/CrkII interactions in cells, which would impact avidity of these complexes rather than the PRM/SH3 interaction.

We therefore propose that high avidity binding between the CrkII$^{NSH3}$ motif and the conserved tandem site PRM on PEAK dimers is a general feature of PEAKs that enhances localization to Fas or other sites of CrkII clustering (Fig. 8a and Supplementary Fig. 7)[30,53]. Whilst we identify an SH2-mediated dimeric form of CrkII$^{FL}$ from over-expression in *E. coli* that illustrates the effect of CrkII clustering, further work will be required to confirm whether this specific CrkII dimer exists in a cellular context. However, as multivalency is a general feature of both CrkII and CrkL signaling complexes, our work has broader implications. A notable example is cell adhesion-mediated CrkII/CrkL interactions with the Cas family of scaffolds (p130Cas or NEDD9), each of which harbor flexible central substrate domains with multiple repeats (15 repeats in p130Cas) of the YxxP motif; a consensus binding sequence for SH2- and PTB-containing proteins such as CrkL, Nck, and SHIP2. Integrin-mediated cell adhesion allows Cas to activate Src/SFKs, which processively phosphorylate the Cas substrate domain following mechanosensitive changes in its accessibility, ultimately enabling multivalent CrkII recruitment[54,55]. In this manner, CrkII cooperates with p130Cas/NEDD9 in dynamic integrin signaling at FAs[56]. Bound and clustered CrkII then recruits other adapters/effectors of other pathways to amplify and diversify the signal, including Dock180/Rac (regulation of migration) and C3G/Rap1 (regulation of cellular morphology and adhesion)[57]. Likewise, AXL-mediated NEDD9 phosphorylation is reported to recruit CrkII and thereby PEAK1, leading to altered FA dynamics in breast cancer cells[30]. This is consistent with roles for p130Cas/NEDD9 in tumor progression through regulating cytoskeletal dynamics[58]. Thus, although not specifically examined in our work, our findings with CrkII likely also have implications for PEAK interactions with CrkL, which interestingly can additionally undergo CSH3-mediated homodimerization in cells[44]. Notably, the SH2-mediated localization of CrkL at FAs is reported to be important for correct co-recruitment of ASAP1 (via the ASAP1 SH3 domain) to FAs in platelets[59]. Further studies would be needed to confirm whether PEAK3 has a role in this process also.

Finally, whilst SH3/PRM interactions do not typically require PTMs for binding, they are by no means static, and we exemplify two mechanisms by which they can be dynamically regulated: via clustering/avidity, as described above; and via accessibility, which can be modulated by the phosphorylation of an adjacent site. We identify a conserved tandem motif (CrkII$^{NSH3}$ site/14-3-3 site) present on all PEAKs (Fig. 5) and define a role for 14-3-3 as a potential negative regulator of PEAK3/Crk interactions at FAs (Fig. 8 and Supplementary Fig. 7a). We demonstrate that PEAK3$^{FL}$ forms a highly stable dimer:dimer complex with 14-3-3, an interaction that prevents stable PEAK3 binding to dimeric CrkII$^{FL}$ (Fig. 5). Our ITC competition studies with PEAK3 tandem peptides confirm that the CrkII$^{NSH3}$ and 14-3-3 directly exhibit negative cooperativity for binding to the PEAK3 tandem site, providing a mechanistic explanation for the observed effect on PEAK3/CrkII interaction with full-length proteins (Figs. 6 and 4d).

PEAK3 appears to be unique from PEAK1/2 in its interaction with 14-3-3 at the tandem motif, displaying ~10-fold higher affinity for 14-3-3 (PEAK3 pS69), compared to either PEAK1 (pT1165) or PEAK2 (pS826). For PEAK3, the pS69 site is sufficient for stable 14-3-3 recruitment, whereas the longer IDRs of PEAK1/2 are predicted to harbor additional 14-3-3 motifs and different mechanisms may exist. For example, the conserved tandem site 14-3-3 motif might act as a phosphodependent secondary (low-affinity) 14-3-3 site to alter PEAK1/2 activity following initial 14-3-3 recruitment at a distal "gatekeeper" (high-affinity) site[60,61].

14-3-3 binding has been reported to provide a regulatory switch in proteins involved in cytoskeletal dynamics (e.g., SSH1L, cortactin and IRSp53) by regulating downstream effector interactions and sub-cellular localization[62–65]. For IRSp53, 14-3-3 binding masks SH3-mediated docking with effectors at lamellipodia and filopodia, terminating IRSp53 signaling[63–65]. As CrkII and Grb2 appear to cooperate in PEAK3 signaling[14], PEAK3-pS69 phosphorylation and 14-3-3 binding may thus provide an analogous regulatory feedback loop to switch on- or off- PEAK3/Crk/Grb2-mediated signaling at FAs, perhaps altering PEAK3 sub-cellular localization (Fig. 8 and Supplementary Fig. 7). Consistent with our findings that S69A mutation of PEAK3 disrupts 14-3-3 binding and leads to a modest increase in association with CrkII, we observe that S69A PEAK exhibits enhanced pY24 phosphorylation (Fig. 7), which is expected to enhance SH2-mediated Grb2 recruitment (Fig. 8). We and others have reported that PEAK3 forms complexes in cells with Grb2/PYK2/ASAP1[14,23]. In THP1 cells in the absence of growth factors, PEAK3 dimerization was reported to result in PYK2-mediated activation of ASAP1 and activation of PI3K/AKT signaling[23]. We propose that the S69 site on PEAK3 may therefore act as a critical regulatory site for PEAK3 activity in these signaling pathways, via phosphorylation by a basophilic serine/threonine kinase, however further studies are required to confirm the relevant kinase/s for the PEAK3 S69 site in cells.

Together our work highlights distinct mechanisms of regulation and signaling outputs possible from homo/heterodimeric PEAK complexes (Supplementary Fig. 7b). We characterize the role of the conserved IDR motif and dimerization/avidity in CrkII binding, features expected to be common to all PEAK complexes. We illuminate additional features of an IDR interaction with Grb2 present only in PEAK3/PEAK1. Lastly, we identify a role for the IDR tandem site in mediating PEAK binding to 14-3-3 proteins. For PEAK3, phosphoregulation at S69 enables formation of a distinct high affinity PEAK3:14-3-3 complex and we propose a mechanistic role in negative regulation of PEAK3/CrkII/Grb2 signaling. This provides a framework to better understand the contribution of intrinsically disordered regions to signaling of dimeric PEAK family pseudokinase scaffolds in healthy cells and cancer.

## Methods

### PEAK multiple sequence alignment (MSA) and SliM analysis

Vertebrate orthologs of human PEAK-family proteins (PEAK3, PEAK1 and PEAK2/PRAG1) were compiled from the NBCI Gene database[66] (calculated by NCBI's Eukaryotic Genome Annotation pipeline) and manually curated to remove sequences annotated by NCBI as low quality. This yielded orthologs for PEAK3 (149 vertebrate orthologs), PEAK1 (301 vertebrate orthologs) and PEAK2 (206 vertebrate orthologs). Multiple sequence alignment for each set of PEAK orthologs and for the full family of all PEAK orthologs was performed using PROMALS3D[67] and visualized by web logo generation using WebLogo3[68]. Short linear motif (SLIM) analysis was performed using the Eukaryotic Linear Motif (ELM) server[69] and Scansite 4.0[70] and manual annotation.

### Synthetic peptides and commercial recombinant protein

All synthetic peptides were purchased from Mimotopes (95% purity).

Five 13-mer peptides encompassing proline-rich motifs from PEAK proteins were synthesized: PEAK3[54–66] (Ac-PLPPPLPKKILTR-NH2); PEAK1[1117–1129] (Ac-MIPPKQPRQPKGA-NH2); PEAK1[1150–1162] (Ac-PTPPPLPKKMIIR-NH2); PEAK2[709–721] (Ac-FSPPPPPPKSRHL-NH2); and PEAK2[809–821] (Ac-QPPPLPQKKIVSR-NH2).

A 7-mer sequence containing the putative Grb2/CrkII SH2 binding motif is common to both human PEAK3 (residues 23–29) and human PEAK1 (residues 1106–1112). Two synthetic 7mer peptides corresponding to this sequence were synthesized, either phosphorylated or non-phosphorylated at the tyrosine residue (PEAK SH2-pY consensus peptide, Ac-T(pY)SNLGQ-NH2; and Ac-TYSNLGQ-NH2; both

N-terminal acetylated, C-terminally amidated)[14]. Two additional longer 16mer peptides were also synthesized spanning the consensus phosphotyrosine motif and additional adjacent non-conserved sequence that might also interact with SH2 domains of Grb2 or CrkII−PEAK3 residues 23–38 pY24 (PEAK3[23–38]-pY24, Ac-T(pY)SNLGQIRAHLLPSK-NH2) and PEAK1 residues 1106–1121 pY1107 (PEAK1[1106–1121]-pY1107, Ac-T(pY)SNLGQSRAAMIPPK-NH2).

Tandem peptides (phosphorylated and non-phosphorylated at the bolded/underlined residue) were also synthesized: PEAK3[tandem] and PEAK3[tandem]-pS69 (residues 54–74; Ac-PLPPPLPKKILTRTQ**S**LPTRR-NH2); PEAK1[tandem] and PEAK1[tandem]-pT1165 (residues 1152–1171; Ac-PPPLPKKMIIRAN**T**EPISKD-NH2); and PEAK2[tandem] and PEAK2[tandem]-pS826 (residues 812–831; Ac-PPPLPQKKIVSRAA**S**SPDGF-NH2).

### Protein production of His-tagged adapter/scaffold proteins

Human gene sequences of adapter/scaffold proteins (full length or truncated versions) were codon optimized for *E. coli* expression and cloned into a pCOLD™-IV vector (Takara) modified to encode an N-terminal 8-His tag and tobacco etch virus (TEV) protease cleavage site for tag removal. Several constructs were generated for this study. For adapter proteins this included: full length Grb2 (Grb2[FL]; UniProt P62993-1, residues 1–217); full length CrkII (CrkII[FL]; UniProt P46108-1, residues 1–330), and truncated forms of CrkII, including a SH2-NSH3 domain construct lacking the CSH3 domain (CrkII[ΔCSH3] residues 1–228) and constructs of the individual CrkII SH2 domain (CrkII[SH2]; UniProt P46108-1, residues 6–120) and CrkII NSH3 domain (CrkII[NSH3]; UniProt P46108-1, residues 134–191). For 14-3-3 isoforms this included: 14-3-3ε (UniProt P62258-1; residues 1–255), 14-3-3η (UniProt Q04917-1; residues 1–246), 14-3-3γ (UniProt P61981-1; residues 1–247) and 14-3-3ς (UniProt P63104-1; residues 1–245). All constructs were verified using Sanger Sequencing (Micromon). Expressions were carried out overnight at 16 °C for 16 to 20 h in *E. coli* C41 (DE3) and proteins were purified using nickel affinity chromatography and size exclusion chromatography (SEC). When required, the His-tagged protein was cleaved following SEC with TEV protease. Cleaved samples were further purified by ion exchange chromatography (IEX) using Mono Q™ 5/50 GL (CrkII[FL], 14-3-3 and Grb2[FL]) columns (Cytiva) (Supplementary Table 1). All protein samples were concentrated, flash frozen and stored at −80 °C.

### Protein production of PEAK proteins

Human PEAK1 (PEAK1[IDR1]; UniProt Q9H792, residues 1082–1746), PEAK2 (PEAK2[IDR1]; UniProt Q86YV5, residues 802–1406) and PEAK3 (PEAK3[FL]; UniProt Q6ZS72, residues 1–473) were codon optimized for insect cell expression and cloned into a modified pFastBac™ Dual vector containing a N-terminal TEV cleavable His-tag under the polyhedrin promoter[35]. Expression was carried out in *Spodoptera frugiperda* (*Sf*21) cells and purification was performed as previously described[35] (Supplementary Table 2). Briefly, cell lysates were clarified by centrifugation at 45,500 × *g* for 1 h at 4 °C and mixed with 0.5% (v/v) of His-tag purification resin (Roche) for purification by nickel affinity chromatography, PEAK proteins were further purified by SEC (HiLoad 16/600 Superdex 200 pg column). For PEAK3[FL], the resulting sample contained a mixture of PEAK3 protein in complex with two proteins identified by mass spectrometry as endogenous insect cell-derived 14-3-3ζ and 14-3-3ε (Supplementary Fig. 4a). PEAK1[IDR1] and the PEAK3[FL]:14-3-3ε,ζ complex were further purified by IEX (Mono Q™ 5/50 GL column). PEAK1[IDR1], PEAK2[IDR1] and the PEAK3[FL]: 14-3-3ε,ζ complex were concentrated to ~5 mg ml⁻¹ and flash frozen in liquid nitrogen for subsequent studies. Human PEAK3 (PEAK3[FL]; UniProt Q6ZS72, residues 1–473) was also cloned into a pCold™ IV vector (Takara) with a N-terminal cleavable His-tag. Expression was carried in *E. coli* C41 (DE3) cells at 16 °C overnight after 0.5 mM IPTG induction. A two-step purification was performed using the methodology described previously[35].

## Recombinant kinase domains for in vitro phosphorylation reactions

Constructs for expression of recombinant Src kinase domain (UniProt P12931, residues 254–536; Addgene plasmid #79700), Abl kinase domain (UniProt P00519-1, residues 229–512; Addgene plasmid #78173) and untagged YopH phosphatase (UniProt P08538-1, residues 164–468; Addgene plasmid #79749) from the Human Kinase Domain Constructs Kit for Automated Bacterial Expression (Addgene plasmid kit #1000000094) were a gift from John Chodera (Memorial Sloan Kettering Cancer Center)[71]. Co-expressions of Src/YopH or Abl/YopH were carried out overnight at 16 °C for 16 to 20 h in *E. coli* C41 (DE3) or *E. coli* Rosetta (DE3), respectively and purified as described using nickel affinity chromatography and SEC, without cleavage of the His-tag. Purified Abl kinase domain was stored at 2 mg ml⁻¹ in 20 mM Tris pH 7.5, 150 mM NaCl, 5% glycerol, 0.5 mM TCEP. The Src kinase domain was further purified by IEX (Mono Q™ 5/50 GL column) and stored at 12 mg ml⁻¹ in 20 mM Tris pH 8.0, 165 mM NaCl, 1 mM TCEP for subsequent studies (Supplementary Table 3).

## Complex studies using analytical SEC and SEC-MALS

SEC analysis of PEAK3$^{FL}$, PEAK2$^{IDR1}$, PEAK1$^{IDR1}$ and CrkII$^{FL}$ dimer/monomer (Figs. 3 and 4 and Supplementary Fig. 3) were performed on a Superdex™ 200 Increase 10/300 GL (GE Healthcare) equilibrated in 20 mM HEPES pH 7.5–8.5, 150 mM NaCl, 1 mM TCEP. For individual analysis, PEAK and CrkII proteins were prepared at 20–50 µM concentration in a total volume of 110 µl. To test the stability of purified CrkII$^{FL}$ dimer and CrkII$^{FL}$ monomer, two 100 µl injections of each of 50 µM CrkII$^{FL}$ dimer and CrkII$^{FL}$ monomer were re-run on SEC, S200 10/300 pre-equilibrated in 20 mM HEPES pH 7.5, 150 mM NaCl, 1 mM TCEP, immediately after thawing and after 45 h incubation at 4 °C. For complex interaction studies, PEAK and CrkII proteins were mixed at a 1:1–2 molar ratio (20–50 µM final concentration) in a total reaction volume of 110 µl, and the complex was pre-equilibrated at room temperature for 25 min and filtered prior to SEC analysis. Elution peak fractions were analyzed by SDS-PAGE and peak elution fractions were pooled and concentrated and flash frozen in liquid nitrogen for subsequent studies.

SEC multi-angle light scattering (SEC-MALS) experiments were performed using a Superdex 200 Increase 10/300 GL column (Cytiva) coupled with a DAWN HELEOS II light scattering detector and an Optilab TrEX refractive index detector (Wyatt Technology). The system was equilibrated in 20 mM HEPES pH 7.5, 150 mM NaCl, 1 mM TCEP and calibrated using bovine serum albumin (2 mg ml⁻¹; Sigma) before analysis of experimental samples. For each experiment, 100 µl of purified pre-equilibrated protein complex (120 µg of PEAK3$^{FL}$: 14-3-3ε,ζ and 170 µg of PEAK2$^{IDR1}$:CrkII$^{FL}$ dimer) was injected onto the column and eluted at a flow rate of 0.5 ml min⁻¹. Experimental data were collected and processed using ASTRA software (Wyatt Technology, v.7.3.19).

## Mass spectrometry (MS)

Tryptic MS phosphosite analysis of recombinant PEAK proteins: Recombinant purified PEAK1$^{IDR1}$, PEAK2$^{IDR1}$ and PEAK3$^{FL}$ were reduced with DTT (Sigma), carbamidomethylated with iodoacetamide (Sigma), and digested with trypsin (Promega). Prior to MS, the peptides were further purified and enriched using OMIX C18 Mini-Bed tips (Agilent Technologies). Using a Dionex UltiMate 3000 RSLCnano system equipped with a Dionex UltiMate 3000 RS autosampler, an Acclaim PepMap RSLC analytical column (75 µm × 50 cm, nanoViper, C18, 2 µm, 100 Å; Thermo Fisher Scientific), and an Acclaim PepMap 100 trap column (100 µm × 2 cm, nanoViper, C18, 5 µm, 100 Å; Thermo Fisher Scientific), the tryptic peptides were separated by increasing concentrations of 0.1% (v/v) formic acid (prepared in 80% (v/v) acetonitrile) at a flow rate of 250 nl min⁻¹ for 50 min and analyzed with a Q-Exactive HF mass spectrometer (Thermo Fisher Scientific). To obtain peptide and phosphopeptide sequence information, the raw files were

searched with Byonic (Protein Metrics, v. 3.1.0) against a human database obtained from UniProt (UP000005640) with the following parameters: carbamidomethylation at cysteine residues was specified as a fixed modification; oxidation at methionine residues and phosphorylation at serine, threonine, or tyrosine residues were set as variable modifications; cleavage site specificity was set at C-terminal residues K and R with semi-specific cleavage specificity with up to two missed cleavages permitted; and precursor and fragment ion mass tolerances were set to 20 ppm. Only peptides identified within a false discovery rate of 1% based on a decoy database were reported.

For in vitro ABL/SRC phosphorylation of PEAKs, CrkII and PTM analysis using tandem MS/MS, purified recombinant PEAK3$^{FL}$ and PEAK1$^{IDR1}$, CrkII$^{FL}$ were phosphorylated in vitro using recombinant Abl and Src. Substrate proteins were incubated at room temperature for 3 h with recombinant Abl or Src kinase domains (5:1 molar ratio of substrate to kinase, 100 µl reaction volume) in 40 mM Tris pH 8.0, 150 mM NaCl, 10 mM MgCl$_2$, 1 mM DTT and 5 mM ATP. Reactions were stopped with 25 mM EDTA. Control samples (no Abl or Src added to reaction) were performed analogously and included in MS analysis. MS analysis was performed as for recombinant PEAK proteins. Peptide and phosphopeptide sequences were obtained in Byonic using the same parameters as the tryptic MS phosphosite analysis.

For 14-3-3 identification, the eluted protein complex from SEC was digested in-solution, following reduction and alkylation, for 16 h, at 37 °C using trypsin. Following digestion, samples were transferred to pre-equilibrated C18 StageTips (2× plugs of 3 M Empore resin, no. 2215) for sample clean-up. The eluates were lyophilized to dryness using a CentriVap (Labconco), before reconstituting in 10 µl 0.1% FA/2% ACN ready for mass spectrometry analysis. Peptides (5 µl) were then separated by reverse-phase chromatography on a C$_{18}$ fused silica column (inner diameter 75 µm, OD 360 µm × 25 cm length, 1.6 µm C$_{18}$ beads) packed into an emitter tip (IonOpticks), using a nano-flow HPLC (M-class, Waters). The HPLC was coupled to a timsTOF Pro (Bruker) equipped with a CaptiveSpray source. Peptides were loaded directly onto the column at a constant flow rate of 400 nl min⁻¹ with buffer A (99.9% Milli-Q water, 0.1% FA) and eluted with a 30-min linear gradient from 2 to 34% buffer B (99.9% ACN, 0.1% FA). The timsTOF Pro was operated in PASEF mode using Compass Hystar 5.1. Settings as follows: Mass Range 100 to 1700 *m/z*, 1/K0 Start 0.6 V s/cm² End 1.6 V s/cm², Ramp time 110.1 ms, Lock Duty Cycle to 100%, Capillary Voltage 1600V, Dry Gas 3 l/min, Dry Temp 180 °C, PASEF settings: 10 MS/MS scans (total cycle time 1.27 s), charge range 0–5, active exclusion for 0.4 min, Scheduling Target intensity 10,000, Intensity threshold 2500, CID collision energy 42 eV. Raw data files were analyzed by MaxQuant software (v. 1.6.17) using the integrated Andromeda search engine. Experiment type was set as TIMS-DDA with no modification to default settings. Data were searched against both Human and Noctuidae Uniprot reference proteomes (downloaded March 2022) and a separate reverse decoy database using a strict trypsin specificity allowing up to 2 missed cleavages. The minimum required peptide length was set to 7 amino acids. Modifications: Carbamidomethylation of Cys was set as a fixed modification, while N-acetylation of proteins and oxidation of Met were set as variable modifications. First search peptide tolerance was set at 10 ppm and main search set at 20 ppm (other settings left as default). Peptide-spectrum match and protein identifications were filtered using a target-decoy approach at an FDR of 1%. Label-free quantification (LFQ) quantification was selected, with a minimum ratio count of 2. Peptide-spectrum match scores and protein identifications were filtered using a target-decoy approach at an FDR of 1%. Average protein LFQ values across replicates were plotted to identify proteins present in the samples.

## Protein crystallization, data collection and processing

Crystallization of the Grb2$^{FL}$:PEAK SH2-pY peptide complex was accomplished based on the previously described method for Grb2$^{FL}$

alone[36], using the hanging-drop vapor diffusion method at 20 °C, and initial crystals optimized by varying pH, streak seeding and adjusting the drop ratio. The protein complex was prepared by concentrating Grb2$^{FL}$ to 16 mg ml$^{-1}$ with 1 mM peptide (diluted from 10 mM stock in water). Crystals used for data collection were obtained in 2.6 M sodium acetate pH 8.0, by mixing the protein/peptide complex with the well solution in a 2:1 ratio. Crystals were cryo-protected in 2.7 M sodium acetate pH 8.0, 10 % (v/v) glycerol, supplemented with 1 mM peptide and flash-frozen in liquid nitrogen. Diffraction data were collected on beamline MX2 at the Australian Synchrotron[72] using a wavelength of 0.9537 Å. Data were integrated by using XDS (v. 20161205)[73], scaled by using XSCALE (v. 20161205), and merged using AIMLESS (v. 0.5.21)[74,75]. The crystals belonged to space group $P\,4_3$ with unit cell parameters $a = b = 89.0$, $c = 94.5$ Å and $\alpha = \beta = \gamma = 90°$, and contained two copies of Grb2$^{FL}$ per asymmetric unit as a closely packed homodimer. The structure was solved by molecular replacement using PHASER (v. 2.8.3)[76] with Grb2$^{FL}$ monomer coordinates derived from the existing Grb2$^{FL}$ structure (PDB entry 1GRI) as the initial search model. Clear unbiased density was observed for the phosphopeptide in the SH2 phosphotyrosine binding site for one copy of Grb2$^{FL}$ in the dimer, whilst no additional density was observed in the SH2 site in the second copy, which appeared to be occluded due to crystal packing. Iterative model building and refinement were performed in COOT (v. 0.9.8.1)[77] and PHENIX (v. 1.20.1-4487)[78], respectively, including a simulated annealing step in the first round of refinement to minimize model bias and manual building of the phosphopeptide. The structure was refined to Rwork and Rfree values of 16.6% and 20.9%, respectively, with all of the residues in Ramachandran allowed regions as validated by MOLPROBITY[79]. The overall structure aligns closely with the earlier published structure for Grb2$^{FL}$ alone (PDB entry 1GRI, 3.1 Å resolution), with the addition of the PEAK1/3 pY-site phosphopeptide and some additional Grb2$^{FL}$ residues able to be modeled due to the higher resolution (2.3 Å resolution).

Crystallization of the 14-3-3ε:PEAK3$^{tandem}$-pS69 peptide complex was achieved by incubating 14-3-3ε (10 mg ml$^{-1}$ final concentration) with the PEAK tandem peptide (0.5 mM final concentration, diluted from 10 mM stock in water), representing a 1.5-fold molar excess of peptide. The complex was subjected to sparse matrix screening in a 96-well plate using the sitting-drop vapor diffusion method at 20 °C and a 1:1 drop ratio of protein and well solution. Diffracting crystals were obtained in 2 M ammonium sulfate, 0.1 M sodium bicine pH 9.35, and 5% (v/v) 2-methyl-2,4,-pentandiol (MPD). Crystals were cryo-protected in 2.2 M ammonium sulfate, 0.02 M sodium bicine pH 9.35, 5% (v/v) MPD, and 20% (v/v) ethylene glycol supplemented with 100 μM peptide and flash-frozen in liquid nitrogen. Data collection and processing was performed as for the Grb2$^{FL}$:PEAK SH2-pY peptide complex. The crystals belonged to space group P $4_3$ with unit cell parameters $a = b = 155.42$, $c = 58.0$ Å and $\alpha = \beta = \gamma = 90°$, and contained four copies of 14-3-3ε monomer per asymmetric unit, arranged as two symmetrical homodimers as anticipated. Molecular replacement, model building and refinement were performed as for the Grb2$^{FL}$:PEAK SH2-pY peptide complex except that the human 14-3-3ε monomer structure (PDB entry 2BR9) was used as the initial search model. Clear unbiased density was observed for the phosphopeptide in the binding site of each 14-3-3ε monomer, although chain A showed the clearest resolved density following building and refinement. The structure was refined to Rwork and Rfree values of 19.2% and 23.3%, respectively, with 99.8% of the residues in Ramachandran allowed regions as validated by MOLPROBITY[79].

The crystal structure of the 14-3-3ε:PEAK1$^{tandem}$-pT1165 complex was obtained by firstly crystallizing 14-3-3ε alone and then soaking the crystals with the PEAK tandem phosphopeptide. Successful conditions identified for the 14-3-3ε:PEAK3$^{tandem}$-pS69 peptide complex were used as the basis for 14-3-3ε crystallization trials using the hanging-drop vapor diffusion method at 20 °C. Crystals of 14-3-3ε (10 mg ml$^{-1}$) were obtained in 1.8 M ammonium sulfate, 0.1 M HEPES pH 7.5, 5% (v/v) MPD, then transferred to well solution supplemented with 0.5 mM phosphopeptide to soak overnight. Crystals were cryo-protected in 2.2 M ammonium sulfate, 0.1 M HEPES pH 7.5, 5% (v/v) MPD, 20% (v/v) ethylene glycol supplemented with 0.5 mM phosphopeptide and flash-frozen in liquid nitrogen. Data collection, molecular replacement, and model building and refinement were performed as for the 14-3-3ε:PEAK3$^{tandem}$-pS69 complex. The crystals belonged to space group $P\,6_2\,2\,2$ with unit cell parameters $a = b = 91.4$, $c = 139.9$ Å and $\alpha = \beta = 90°$, $\gamma = 120°$ and contained one 14-3-3ε monomer per asymmetric unit. This particular crystal form represents a close crystal packing of 14-3-3ε as a monomer, in which residues 1–32 that would typically mediate the 14-3-3ε dimer interface are not resolved, as the crystal packing would require them to be at least partially unfolded. The remainder of the 14-3-3ε protein core aligns closely with the structure of dimeric 14-3-3ε (as for the 14-3-3ε:PEAK3$^{tandem}$-pS69 complex). Notably, the peptide binding groove is accessible and amenable to soaking and clear unbiased density for the PEAK phosphopeptide was observed. The final structure was refined to Rwork and Rfree values of 22.8% and 28.8%, respectively, with 99.5% of the residues in Ramachandran allowed regions as validated by MOLPROBITY[79].

Crystallization, soaking, data collection and processing, and model building and refinement of 14-3-3ε:PEAK2$^{tandem}$-pS826 complex were performed identically as for the 14-3-3ε:PEAK1$^{tandem}$-pT1165 complex, but using the PEAK2$^{tandem}$-pS826 phosphopeptide. The crystals belonged to space group $P\,6_2\,2\,2$ with unit cell parameters $a = b = 92.1$, $c = 137.4$ Å and $\alpha = \beta = 90°$, $\gamma = 120°$, and contained one 14-3-3ε monomer per asymmetric unit, in the same close packing of the 14-3-3ε monomer. The final structure was refined to Rwork and Rfree values of 24.6% and 30.4%, respectively, with all the residues in Ramachandran allowed regions as validated by MOLPROBITY[79].

## AlphaFold2 structure prediction of PEAK3$^{FL}$ homodimer
The ColabFold Google Colab notebook called AlphaFold2_ complexes[80,81] was used to predict the structure of dimeric human PEAK3$^{FL}$. The predicted model with the highest lDDT score is shown.

## Surface plasmon resonance (SPR) binding assays
SPR binding studies of PEAK PRM peptides to CrkII$^{NSH3}$ were performed using a Biacore S200 Instrument (Cytiva). Purified CrkII$^{NSH3}$ was diluted to 5 μg ml$^{-1}$ in 10 mM sodium acetate pH 4.0 and amine coupled at 25 °C to a Series S CM5 sensor chip, in HBS-N running buffer (20 mM HEPES pH 7.4, 150 mM NaCl) to a final immobilization level of 1100–1400 response units (RU), followed by surface deactivation using 1 M ethanolamine. A blank activation/deactivation was used for the reference surface. Peptide binding studies were performed at 20 °C in HBS-TP running buffer (20 mM HEPES pH 7.4, 150 mM NaCl, 1 mM TCEP, 0.005% (v/v) Tween-P20). Peptides were first prepared as 10 mM stocks in water: PEAK PRM peptides: PEAK3$^{54-66}$, PEAK1$^{1150-1162}$, PEAK2$^{709-721}$, PEAK2$^{809-821}$; or PEAK tandem motif peptides: PEAK3$^{tandem, 54-74}$, PEAK3$^{tandem,54-74}$-pS69, PEAK1$^{tandem,1152-1171}$, PEAK1$^{tandem,1152-1171}$-pT1165, PEAK2$^{tandem,812-831}$, and PEAK2$^{tandem,812-831}$-pS826, phosphorylated or non-phosphorylated in the 14-3-3 motif. Peptide stocks were further diluted to in running buffer to 10 or 20 μM and prepared as a 11-point concentration series (2-fold serial dilution, 100 μM–100 nM for PEAK PRM peptides, or 20 μM–20 nM for PEAK tandem motif peptides). Samples were injected in a multi-cycle run (flow rate 30 μl min$^{-1}$, contact time of 60 s, dissociation 120 s) without regeneration. Sensorgrams were double referenced, and steady-state binding data fitted using a 1:1 binding model using Biacore S200 Evaluation Software (Cytiva, v. 1.1). Representative sensorgrams and fitted dissociation constant ($K_D$) values, depicted as mean ± S.E.M. ($n \geq 3$ independent experiments) or mean ± S.D. ($n = 2$), are shown in Supplementary Data 1–6 (SPR data).

SPR binding studies of PEAK tandem peptides to 14-3-3 isoforms were performed using a Biacore S200 Instrument (Cytiva).

Immobilization of 14-3-3 isoforms (14-3-3γ, 14-3-3ε, 14-3-3σ, and 14-3-3η; diluted to 10 μg ml⁻¹ in 10 mM sodium acetate pH 4.0) was performed as described for CrkII^NSH3 at 20 °C to a final immobilization level 1800–2900 RU. Binding studies were run using PEAK tandem peptides (11-point concentration series, 2-fold serial dilution; 10 μM–10 nM for PEAK3^tandem-pS69, 20 μM–20 nM for all other peptides). SPR binding experiments and steady-state data analysis were performed as described for CrkII^NSH3. Representative sensorgrams and fitted dissociation constant ($K_D$) values, depicted as mean ± S.E.M. ($n \geq 3$ independent experiments), are shown in Supplementary Data 1–6 (SPR data).

PEAK proteins and adapters proteins (full length or individual domains) were biotinylated using EZ-Link™ NHS-PEG4-Biotin (Thermo Fisher) at a 1:1 molar ratio (100 μM) for 1 h at room temperature, then excess biotin was removed by buffer exchange using a Zeba™ Spin Desalting Column (Thermo Fisher) into HBS-T buffer (20 mM HEPES pH 7.4, 150 mM NaCl, 1 mM TCEP).

SPR binding studies of PEAK pY phosphopeptides to CrkII/Grb2 and sub-domains. Binding studies were performed using a Biacore 8K+ Instrument (Cytiva) and analyzed using Biacore Insight Evaluation Software (v. 3.0.12.15655). Biotinylated adapter proteins (Grb2^FL, Grb2^SH2, CrkII^SH2, CrkII^FL monomer, CrkII^FL dimer) were immobilized using a biotin CAPture kit (Cytiva) in HBS-TP running buffer at 20 °C. Binding studies and steady-state data analysis using PEAK pY peptides (consensus SH2-pY peptide; PEAK3^23–38-pY^24 peptide and PEAK1^1106–1121-pY^1107 peptide) were performed as described for CrkII^NSH3 with the following modifications: HBS-PT running buffer was supplemented with 2% DMSO to reduce nonspecific interactions and an additional regeneration step (60 s injection of 1 M NaCl) was included between cycles. Representative sensorgrams and fitted dissociation constant ($K_D$) values, depicted as mean ± S.E.M. ($n \geq 3$ independent experiments), are shown in Supplementary Data 1–6 (SPR data).

SPR binding studies of PEAK proteins to full length CrkII/Grb2 and sub-domains were performed using a Biacore 8K+ Instrument (Cytiva) and analyzed using Biacore Insight Evaluation Software (v. 3.0.12.15655). Biotinylated PEAK3^FL (14-3-3 complex), PEAK1^IDR1 and PEAK2^IDR1 purified from insect cells and PEAK3^FL (no bound 14-3-3) purified from *E.coli*, were phosphorylated using recombinant Src kinase as previously described. Biotinylated PEAK proteins (with or without Src phosphorylation; 500 nM in HBS-TP) were immobilized using a biotin CAPture kit (Cytiva) in HBS-TP running buffer at 20 °C. Binding studies and steady-state data analysis were performed as described for CrkII^NSH3 with the inclusion of an additional surface regeneration step (60 s injection of 1 M NaCl) between cycles. For Grb2^FL, Grb2^SH2, CrkII^SH2, CrkII^FL monomer and CrkII^FL dimer, to reduce nonspecific interactions the HBS-TP running buffer was supplemented with additional NaCl to a final concentration of 500 mM.

Binding studies for 14-3-3γ were undertaken in the same way in multi cycle mode (8 point, 3-fold serial dilution series, 0.46–1000 nM, 60 s contact time, 300 s dissociation). Of the PEAKs, only PEAK3^FL purified from insect cells was observed to bind recombinant 14-3-3γ, irrespective of Src phosphorylation of PEAK3^FL. This is consistent with the biotinylated PEAK3^FL purified from insect cells being prepared from a complex bound to endogenous 14-3-3 and phosphorylation of this sample at S69 in the tandem site by MS analysis. The availability of this site to reversibly bind recombinant 14-3-3γ in these SPR experiments indicates that, under the constant flow conditions of the SPR experiment, the majority of insect cell-derived 14-3-3 was able to dissociate from immobilized PEAK3^FL. Some insect-cell derived 14-3-3 may remain captured on the sensor surface, however this did not appear to significantly influence binding interactions, as similar data were obtained for PEAK3^FL expressed in *E.coli* that did not bind 14-3-3. An accurate $K_D$ could not be determined for 14-3-3γ to insect cell derived PEAK3^FL, due to the bivalence of this interaction that does not fit a 1:1 kinetic model. However, fitting using a bivalent analyte model suggests

a high-affinity interaction with likely $K_D \ll 1 \mu M$, consistent with the qualitatively slow dissociation kinetics observed.

Representative sensorgrams and fitted dissociation constant ($K_D$) values, depicted as mean ± S.D. ($n = 2$ independent experiments) or ±S.E.M. ($n \geq 3$ independent experiments), are shown in the Supplementary Information and full SPR data can be found in Supplementary Data 1–6.

### Isothermal titration calorimetry (ITC) binding experiments and cooperativity analysis

ITC binding experiments were conducted using a MicroCal iTC200 instrument (Malvern Instruments). Titrations were conducted at 25 °C in reverse mode (peptide in cell, protein in syringe) and consisted of 19 injections of 2 μl peptide solution at a rate of 2 s μl⁻¹ at 90 s time intervals, whilst stirring at 1000 rpm. An initial injection of 0.4 μl protein was made and discarded during data analysis. Proteins (14-3-3γ or CrkII^NSH3) were first dialyzed overnight at 4 °C into ITC buffer (20 mM HEPES pH 7.4, 150 mM NaCl, 1 mM TCEP), filtered and degassed under vacuum. Peptides (PEAK3^tandem or PEAK3^tandempS69) were diluted into filtered, degassed ITC buffer from 10 mM stock solutions prepared in water. For the binary experiment of CrkII^NSH3 binding to PEAK3^tandem peptide, CrkII^NSH3 (140 μM, in the syringe) was titrated into PEAK3^tandem peptide (20 μM, in the cell). For binding studies to examine cooperativity, sequential ITC titrations were performed according to the following general procedure: In the first titration (binary), 14-3-3γ (200 μM, in the syringe) or a buffer control were titrated into PEAK3^tandem or PEAK3^tandempS69 peptide (20 μM, in the cell). Following the first titration, excess solution was removed from the cell, and the syringe was washed and dried. In the second titration (ternary), CrkII^NSH3 (400 μM, in the syringe) was titrated into the peptide complex remaining in the cell (16.8 μM). The concentration of saturated peptide complex in the cell (C) after the first titration (16.8 μM), was calculated using the equation: $C = (C_0 \cdot V_{cell})/(V_{cell} + V_{inj})$ where $C_0$ is the initial peptide concentration (20 μM), $V_{cell}$ is the volume of the cell (200.1 μl) and $V_{inj}$ is the volume of titrant injected during the first titration (38.4 μl). All data were fitted to a single binding site model to obtain the stoichiometry ($N$), dissociation constant ($K_D$) and the enthalpy of binding ($\Delta H$), using MicroCal PEAQ-ITC Analysis software (v.1.41). The reported values are the mean ± S.E.M. ($N > 3$) from independent measurements (Supplementary Data 5).

### Cellular studies of PEAK interactions (HEK293/MCF-10A cells): antibodies and tissue culture

The following antibodies were obtained commercially: anti-HA (Cell Signaling Technology, catalog no. 3724), anti-Flag (Sigma-Aldrich, catalog no. F1804), anti-14-3-3 (Santa Cruz Biotechnology, catalog no. sc-1657), anti-Grb2 (Cell Signaling Technology, catalog no. 3972S), anti-CrkII (Santa Cruz Biotechnology, catalog no. sc-289), p-Tyr (Cell Signaling Technology, catalog no. 8954S), β-actin (Santa Cruz Biotechnology, catalog no. sc-69879). HEK293 (ATCC, CRL-1573) and MCF10A EcoR cells (gift from D. Lynch and J. Brugge, Harvard Medical School) were maintained as previously described[14]. MCF-10A EcoR cells expressing WT or mutated PEAK3 were generated by retrovirally mediated transduction.

### Plasmids

Codon-optimized cDNAs encoding N-terminal HA-tagged or Flag-tagged WT PEAK3^FL and HA-tagged PEAK3^FL mutants were synthesized as previously described[14]. Plasmid transfections in HEK293 cells were performed using Lipofectamine 3000 (Life Technologies, catalog no. L3000015) according to the manufacturer's instructions.

### Cell lysis and immunoprecipitation

Control HEK293 cells or HEK293 cells singly expressing HA-tagged PEAK3^FL WT or HA-tagged PEAK3^FL-S69A mutant, or co-expressing

Flag-tagged PEAK3[FL] WT with various HA-tagged PEAK3[FL] dimerization mutants (PEAK3[FL]-S69A, -L146E, -A436E, or -C453E) were harvested after 48 h post transfection. Cell lysates for co-immunoprecipitation were prepared using standard lysis buffers[82]. MCF-10A cells or cells expressing HA-tagged PEAK3 were cultured in −EGF medium (DMEM/F-12, Gibco) supplemented with hydrocortisone (0.5 µg/ml; Sigma-Aldrich), cholera toxin (100 ng/ml; Sigma-Aldrich), bovine insulin (10 µg/ml; Sigma-Aldrich), and 2% horse serum overnight and then were harvested. Cell lysates for denatured immunoprecipitation were prepared as previously described[14].

Cell lysates were incubated with anti-HA affinity-agarose beads (Sigma-Aldrich, catalog no. E6679) at 4 °C overnight on a shaking platform. After extensive washing with ice-cold lysis buffer, the immune complexes were eluted with SDS−polyacrylamide gel electrophoresis loading buffer and subjected to Western blotting analysis (Dilutions 1:2000 for anti-HA and anti-Flag; 1:1000 for anti-14-3-3, anti-CrkII, anti-Grb2, anti-Tyr; 1:5000 for anti-actin antibodies). Uncropped gels in source data.

## Cell motility assay
MCF-10A cells were seeded in 12-well plates at $1.5 \times 10^4$ cells per well in duplicate and cultured overnight. Then, the cells were cultured in MCF-10A culture medium with mitomycin C (1 µg/ml; Sigma-Aldrich, catalog no. M4287) to prevent cell division. Images were photographed at eight random positions for each condition. Images of each position were taken every 20 min for 24 h using a Leica DFC9000 with Leica Application Suite X software (version: 3.7.5.24914). Cell motility was measured using Fiji (ImageJ) software (version: 2.0.0).

## Illustrations
All illustrations were generated using PyMOL (v. 2.5.4), UCSF ChimeraX (v. 1.5)[83], GraphPad Prism (v. 9.5.1), Microsoft Excel, Adobe Illustrator and InDesign (v.18.2.1).

## Reporting summary
Further information on research design is available in the Nature Portfolio Reporting Summary linked to this article.

## Data availability
Coordinates and structure factors for the X-ray crystal structures have been deposited in the PDB with accession codes 8DGO (Grb2[FL]:PEAK SH2-pY peptide), 8DGP (14-3-3ε:PEAK3[tandem]-pS69), 8DGM (14-3-3ε:PEAK1[tandem]-pT1165) and 8DGN (14-3-3ε: PEAK2[tandem]-pS826); and existing PDB structures 1GRI (Grb2[FL]), 5IH2 (CrkII-NSH3:Abl-758 peptide) and 2BR9 (14-3-3ε). Coordinates for AlphaFold2 (AF2) models are available from the AlphaFold Protein Structure Database Q6ZS72 (human PEAK3) and Q9H792 (human PEAK1). Coordinates for the human PEAK3[SHED-PsK] dimer prepared from AlphaFold2 multimer modeling, annotated mass spectra and additional source data are provided in the Source Data file. All other data are available from the corresponding author on request. Source data are provided with this paper.

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

## Acknowledgements

We thank the scientific and technical assistance of Monash Proteomics and Metabolomics, Monash University, Victoria, Australia and specifically Dr David Steer, and WEHI Proteomics Facility. We would like to thank the staff at the Bio21 C3 Collaborative Crystallization Centre where all initial crystallization experiments were performed and the beamline staff at the Australian Synchrotron where diffraction data were collected. This research was undertaken in part using the MX2 beamline at the Australian Synchrotron, part of ANSTO, and made use of the Australian Cancer Research Foundation (ACRF) detector. We are grateful to the National Health and Medical Research Council (APP1144149) and Australian Research Council (DP190103672 and DP220103638) for Grant support with additional support from the NHMRC IRIISS (9000719) Operational Infrastructure Support Program provided by the Victorian Government, Australia and from the Australian Cancer Research Foundation (to M.J.R., M.G.S., W.D., J.M.H., A.K., L.Y.L., O.P., I.S.L.). J.M.H. was supported by an NHMRC Investigator Grant (2008096). We acknowledge Melbourne Research Scholarship support for A.K. from the University of Melbourne. The contents of this published material are solely the responsibility of the individual authors and do not reflect the views of the NHMRC or funding partners.

## Author contributions

M.J.R., M.G.S., W.D., O.P. and A.K. designed, generated and biochemically characterized proteins and their complexes. M.J.R. and M.G.S. undertook crystallization experiments and M.J.R. solved crystal structures. M.J.R. and A.K. designed experiments and generated SPR and ITC data. T.R.C., O.P., M.J.R. and B.C.L. conducted and analyzed SEC-MALS experiments. M.G.S., M.J.R., O.P., T.A.D. and L.Y.L. generated mass spectrometry data. J.H., X.M. and R.J.D. designed, undertook, and analyzed cell-based experiments. M.J.R., M.G.S., J.M.H., O.P., R.J.D. and I.S.L. analyzed all the data and wrote the manuscript with input from all authors.

## Competing interests

The authors declare no competing interests.
