## [Peer Review File · Nature Communications]

REVIEWER COMMENTS

Reviewer #1 (Remarks to the Author):

The authors study the molecular mechanisms of interaction of the adaptor proteins Grb2 and CrkII, as well as the 14-3-3 scaffold protein with PEA3. PEA3 is a pseudokinase with important scaffolding functions in a variety of cell biological processes and misregulated expression in different tumours. While the two other PEA family members (PEA1 and PEA2) were structurally characterized recently, the structure and protein-protein interaction of PEA3 are much less well studied. A careful bioinformatics analysis identified SLiMs in the N-terminal region of PEA3: a pY consensus motif for the Grb2 SH2 domain and in close proximity a tandem motif comprised of an SH3 binding motif for CRKII and a 14-3-3 binding motif. The manuscript focuses on deciphering the molecular details of these interactions using biochemical, biophysical and structural methods. The authors first present evidence for phosphorylation of the Grb2 SH2 motif by Src (but not Abl) kinases and co-crystallized this pY peptide with full-length Grb2, which the authors previously showed to bind with low micromolar affinity by ITC. The remainder of the paper is focused on studying the interactions of the tandem motif with the CrkII SH3 domain and 14-3-3. First, peptides with the SH3 binding motif (from PEA3 and PEA1) were shown to bind with low micromolar affinity to the CRKII N-SH3 and robust complex formation of dimeric CRKII (but not the monomeric form of the protein) to a PEA2 construct containing the tandem motif was shown. Next, phosphorylation of Ser-69 in the tandem motif is shown to be the interaction site for 14-3-3 and crystal structures of 14-3-3 epsilon with pS-tandem motif peptides of all three PEAs were solved at 2.7-3.2 Å resolution. To decipher the interplay of the CRKII and 14-3-3 binding sites, ITC experiments are presented demonstrating that 14-3-3 binding reduces affinity for CRKII binding to the adjacent site by ~5-fold.

Overall, this is a rigorously conducted study that deciphers the molecular details of the interactions of the adaptor/scaffold proteins Grb2, CrkII and 14-3-3 with the PEA pseudokinases. The results are clearly presented and appropriately interpreted. The complex interactions of PEA kinase that are deciphered in the manuscript constitute important progress for our understanding of PEA biology. Still, some points should be addressed.

Major points:

1. Why were the SEC experiments in Fig 3c/d not done with PEA3, but only with PEA1 and PEA2? In particular, as the rest of the paper is focused on PEA3. As the observed competition of 14-3-3 binding and CRKII binding to the tandem motif of PEA3 is the most intriguing and novel finding of the paper, it would be very important to demonstrate the interaction of CRKII with PEA3 constructs (such as PEA3-FL or PEA3-IDR1) and not only with peptides.
2. The Grb2 SH2 part (Fig 2) seems a bit disconnected from the rest of the paper and is not entirely novel, as this interaction was previously characterized (see ref. 14). Therefore, it could be moved to SI.

3. Full results of ITC data need to be reported. Besides the K_d , N , ΔH and ΔS need to be reported. All values (K_d , N , ΔH and ΔS) should be reported for all independent replicates instead of only reporting the mean \pm SEM.

Minor points:

1. Fig 3c/d and 4a: SDS-PAGE gels are not properly labeled. It is unclear which SEC fraction/elution volume was loaded in which lane. Furthermore, the molecular weight markers are not labeled.
2. Fig 2a and line 156-158: MS data should be shown in SI. It is currently 'hidden' in source data without reference in the results.

Reviewer #2 (Remarks to the Author):

Review of Roy, et. al., 2022, "Structural insights into regulation of the PEAK3 pseudokinase scaffold by 14-3-3"

In this paper, Roy et. al biochemically and structurally characterize the interactions of the pseudokinases PEAK1, PEAK2, and PEAK3 with their binding partners CrkII, Grb2, and 14-3-3. The authors conducted a bioinformatic analysis to identify several sequence motifs in PEAK3 that mediate these interactions and solve crystal structures of a peptide derived from PEAK1/PEAK3 bound to the Grb2 SH2 domain, as well as phosphopeptides from all three PEAK family members bound to 14-3-3. This study also uses biochemical and biophysical studies to analyze the similarities and differences in the way each of the PEAK pseudokinases interacts with CrkII and 14-3-3, as well as the stoichiometry of these interactions and competition between these proteins for binding to PEAK3. Overall, this study reveals some new insights into the molecular basis for signaling by the PEAK pseudokinases, but falls short of advancing the field enough to merit publication in this journal.

While much effort likely went into the four crystal structures in this study, the data collection and refinement statistics for which are quite good, suggesting the structures are of high quality, unfortunately, these structures do not provide much new insight into the interactions with Grb2 SH2 or 14-3-3 beyond what would be expected based on the many previously published structures of complexes of SH2 domains bound to pTyr containing peptides or 14-3-3 proteins bound to pSer containing peptides. There are plenty of such structures already published (for example, >150 1433 –

peptide complexes in the PDB, maybe lesser number of SH2domain-prptide complexes, but there are plenty). Structure of PEAK3-FL bound to SH2 domain from CrkII or PEAK3-FL-bound to 1433 may reveal extended interactions that beyond what is predicted from domain / peptide analyses. There are several examples of these, so the more relevant context is missing here. Moreover, the electron density in Fig 2d for the peptide in the Grb2 SH2 structure appears relatively weak, with only the phosphate and some of the backbone having strong density. The authors speculate that the predicted helix adjacent to the pTyr site is important for binding, but have no experimental data to support this. A crystal structure of a longer peptide including both the helix and pTyr site bound to Grb2 SH2 would elevate the work to some extent, corroborating their computational analysis, including experiments like testing whether mutagenesis of residues in the helix weakens binding.

The observations that CrkII homodimerizes and that only homodimers, and not monomers, interact with the PEAK pseudokinases is very intriguing, but the physiological relevance of the dimer is unclear. It is interesting that only ~10% of the CrkII protein dimerizes. Since the authors did not find any differences between the monomer and dimer protein via MS, presumably the monomer and dimer exist in an equilibrium. It would strengthen the author's hypothesis if they reinjected the CrkII dimer complex or CrkII monomer complex onto SEC to see if the CrkII dimer or monomer remain intact or sample then re-equilibrates into monomer:dimer as seen in the manuscript – of course, this experiment should be done with the same "amount" of the protein injected into SEC. The authors suggest that CrkII might be forming a domain swapped dimer, as has been observed in crystal structures of other SH2 domains. Based on structures of these other domain-swapped SH2 dimers, it might be possible to disrupt CrkII dimerization through mutagenesis. A monomerizing mutation would enable testing of the relevance of CrkII dimerization in cells, and establishing the physiological relevance of CrkII dimerization would increase the impact of this work.

The authors also present biophysical data demonstrating that CrkII and 14-3-3 compete for binding to PEAK3, which has interesting implications for regulation of the PEAK3/CrkII interaction. While the biophysical data supporting this is strong, the cellular IP data is much weaker. For Extended Data Fig 4b, the authors do admit the difference in band intensity for pulldown of CrkII between the WT and S69A mutant is modest. This would be more convincing if there were data for a cellular assay measuring signaling downstream of CrkII that showed a difference when PEAK3 WT or S69A was expressed.

Other comments:

- 1) For the kinase reaction in which Src was shown to phosphorylate SH2 motif on PEAK3, was the reaction specific only to that specific Tyrosines mentioned or was it heterogenous?
- 2) Why not utilize the extended peptide for the SH2-domain binding region (the one with secondary structure predicted from Alpha-fold) in the crystallization trials?

3) Page 6, towards the end there is a lot of speculation without much data and distracts from the rest of the study.

4) Does CrkII NSH3 domain bind to the tandem peptide containing the PRM and 1433 motif in presence of 1433 in the context of full length PEAK3 or the peptide itself?

5) Page 9 – PEAK3:14-3-3 for a stable dimer-dimer heterocomplex – this is a well written section and very relevant and interesting data. The authors should consider obtaining a structure of this complex, with using cryoEM or by co-overexpressing PEAK3-FL with a human 14-3-3 isoform and perform tandem purification with different affinity tags and isolate a homogenous complex for crystallography, potentially.

Reviewer #3 (Remarks to the Author):

The study attempts to build a more comprehensive model of complexes formed by the PEAK proteins, and provides structural and biophysical analyses which allow extrapolation of how these complexes occur. The work clearly enhances the understanding of PEAK proteins and provides new information on their interactions, but there are a number of areas in which I believe the study could be made more robust. These are discussed below.

The crystallographic analysis of Grb2 is clear, and although the characteristic turn in the peptide is obviously revealed they may consider showing the refined 2Fo-Fc/Fo-Fc maps in addition to their unbiased Fo-Fc omit map of the bound PEAK peptide bound. In contrast, the HADDOCK modeling discussed on page 6 concludes that a new conserved surface is created, but this is based on modeling a model's interaction with their structure. This HADDOCK analysis should be removed as too speculative unless it can be supported by experimental evidence such as ITC to demonstrate the extended predicted interaction enhances selectivity for Grb2 over CrkII (i.e. better than the 2.6 μ M vs 7.8 μ M Kds which they previously published for the short 7-mer peptide).

The SH3 analysis of CrkII with PEAK provides a clear description of this interaction, and it is indeed striking that the dimeric CrkII-FL elutes with the dimeric PEAK by SEC. For completeness, they may wish to include the SEC-MALS for full-length CrkII in Extended Fig 3e and an analysis of the affinity of the Grb2 SH3 domain for the PxxP motif. They may also consider the possibility that the tyrosine phosphorylation at the SH2 motif is proximal (~30 residues) to the PxxP motif, so a single CrkII (or indeed Grb2) protein should be able to engage both the pY and PxxP motifs with potentially significantly tighter binding and avidity than the observed dimer-dimer SH3-mediated complex. This would also impact both their pY-SH2 analysis and 14-3-3 cooperativity analysis.

The discovery of the 14-3-3 binding is indeed interesting, and another nice example of ‘fishing’ out 14-3-3 from baculovirus. The crystal structures of 14-3-3/PEAK should be supported by unbiased Fo-Fc omit maps and refined 2Fo-Fc and Fo-Fc maps for all three bound peptides, and Table 1 should present the B-factors for each chain (14-3-3 and PEAK peptides) and should clearly state the peptide residues which are built in each peptide chain. SPR data and traces for the tandem phosphopeptides binding to 14-3-3 isoforms should be presented in an extended figure (currently these data are presented as a heatmap). Additionally, the gaps in the sequence alignment (Figure 1b) for PEAK1 and PEAK3 are in the wrong places; RTQSLP, RANTEP and RAASSP should all align directly above one another.

The titrations presented in Figure 6 are a nice series of experiments. They might briefly discuss why they used 14-3-3 gamma rather than 14-3-3 epsilon which they used for the crystallography. They also should include an extended table of their ITC parameters (n , dS , dH etc) for each run. Their conclusion that there is negative cooperativity between 14-3-3 binding to PEAK3 and CrkII binding would be better supported by an SEC analysis of S69A PEAK3 interaction with dimeric CrkII-FL (as per Figure 4d), where based on their data the S69A mutant would be expected to interact with CrkII dimer in a manner similar to PEAK1 and PEAK2.

Minor –

line 263 – Extended Data Fig 4a refers to wrong panel (should be Fig 4a)

NCOMMS-22-34485A – Response to Reviewers comments

Reviewer #1 (Remarks to the Author):

The authors study the molecular mechanisms of interaction of the adaptor proteins Grb2 and CrkII, as well as the 14-3-3 scaffold protein with PEA3. PEA3 is a pseudokinase with important scaffolding functions in a variety of cell biological processes and misregulated expression in different tumours. While the two other PEA family members (PEA1 and PEA2) were structurally characterized recently, the structure and protein-protein interaction of PEA3 are much less well studied. A careful bioinformatics analysis identified SLiMs in the N-terminal region of PEA3: a pY consensus motif for the Grb2 SH2 domain and in close proximity a tandem motif comprised of an SH3 binding motif for CRKII and a 14-3-3 binding motif. The manuscript focuses on deciphering the molecular details of these interactions using biochemical, biophysical and structural methods. The authors first present evidence for phosphorylation of the Grb2 SH2 motif by Src (but not Abl) kinases and co-crystallized this pY peptide with full-length Grb2, which the authors previously showed to bind with low micromolar affinity by ITC. The remainder of the paper is focused on studying the interactions of the tandem motif with the CrkII SH3 domain and 14-3-3. First, peptides with the SH3 binding motif (from PEA3 and PEA1) were shown to bind with low micromolar affinity to the CRKII N-SH3 and robust complex formation of dimeric CRKII (but not the monomeric form of the protein) to a PEA2 construct containing the tandem motif was shown. Next, phosphorylation of Ser-69 in the tandem motif is shown to be the interaction site for 14-3-3 and crystal structures of 14-3-3 epsilon with pS-tandem motif peptides of all three PEAs were solved at 2.7-3.2 Å resolution. To decipher the interplay of the CRKII and 14-3-3 binding sites, ITC experiments are presented demonstrating that 14-3-3 binding reduces affinity for CRKII binding to the adjacent site by ~5-fold.

Overall, this is rigorously conducted study that deciphers the molecular details of the interactions of the adaptor/scaffold proteins Grb2, CrkII and 14-3-3 with the PEA pseudokinases. The results are clearly presented and appropriately interpreted. The complex interactions of PEA kinase that are deciphered in the manuscript constitute important progress for our understanding of PEA biology. Still, some points should be addressed.

We thank the reviewer for the constructive comments and have addressed them point by point below.

Major points:

1. Why were the SEC experiments in Fig 3c/d not done with PEA3, but only with PEA1 and PEA2? In particular, as the rest of the paper is focused on PEA3. As the observed competition of 14-3-3 binding and CRKII binding to the tandem motif of PEA3 is the most intriguing and novel finding of the paper, it would be very important to demonstrate the interaction of CRKII with PEA3 constructs (such as PEA3-FL or PEA3-IDR1) and not only with peptides.

The SEC experiments were done with PEA3 as per Fig 3c/d and can be found in Fig 4d. However, as mentioned in the text line 264, PEA3 was purified from insect cells as a stable heterodimer with 14-3-3, and when PEA3-FL is bound to 14-3-3, it does not complex with CrkII (Fig 4d). At the time we wrote the paper we were not able to purify PEA3 –FL in insect cells without 14-3-3 bound and attempts to purify PEA3-S69A mutant failed, limiting our ability to demonstrate the interaction between PEA3-FL and CrkII.

However, to address the reviewer's point and demonstrate the interaction between PEA3-FL and CrkII, and confirm the interaction seen by ITC using PEA3-IDR1 (residue 54-74) (Fig 6), we used the following approaches:

- We first tried to express and purify PEAK3-FL from *E. coli* and succeeded in obtaining a small amount of PEAK3-FL unbound to 14-3-3 (see SDS-Page from SEC elution demonstrating that PEAK3-FL is not complexed with 14-3-3 when purified from *E. coli*). Identity of the protein was confirmed by mass spectrometry. This quantity of PEAK3-FL from *E. coli* was insufficient for SEC studies but was sufficient for SPR immobilisation to conduct similar studies by SPR.

- We then conducted comprehensive SPR binding studies using full-length proteins. See details below.
- We then conducted a comprehensive SPR binding studies using immobilised biotinylated full length or Nterminally truncated recombinant PEAKs (insect cell expressed human PEAK1^{IDR1}, PEAK2^{IDR1}, PEAK3^{FL}-14-3-3 complex and bacterially expressed PEAK3^{FL}) to measure the affinity of interactions with full length interactors and relevant subdomains (14-3-3 γ , CrkII^{FL} monomer or dimer, Grb2^{FL}), both with or without prior phosphorylation of PEAKs by Src kinase (see Fig. 8a and associated tables in Supplementary section: Table Ext7 and Table Ext7 SPR data). **These SPR studies confirmed the key interactions previously mapped using individual peptides.** Notably, whilst insect cell expressed PEAK3^{FL} (S69 phosphorylated) was purified and immobilised as a complex with insect derived 14-3-3 (refer to experimental methods), it showed strong and reversible binding to injected recombinant human 14-3-3 γ , confirming that endogenous insect 14-3-3 had dissociated from PEAK3^{FL} under the continuous flow conditions in SPR experiment. Consistent with PEAK phosphorylation state and measured affinity of individual tandem peptides (Extended Fig. 5c), **only insect cell expressed human PEAK3^{FL} (and no other PEAKs tested, including human PEAK3^{FL} purified from *E.coli*) bound strongly to 14-3-3 γ .** Additionally, **phosphorylation of PEAKs by Src was required for Grb2^{SH2} and CrkII^{SH2} to bind, with Grb2^{SH2} demonstrating tightest binding to PEAK3^{FL}, whilst CrkII^{SH2} bound most tightly to PEAK1^{IDR1} (Fig 8b) consistent with SPR results using individual pY phosphopeptides from the SH2 motif (Fig. 2e).** As for studies using PEAK tandem site proline rich motif peptides, the affinity of CrkII^{NSH3} towards PEAK dimers was found to be relatively similar across all PEAKs (K_D in low micromolar range) irrespective of Src phosphorylation state, reinforcing the understanding of the NSH3/PRM interaction as a primary PEAK-CrkII recruitment site (Fig. 8a,b). The data also reveal contributions of multiple interaction domains to avidity in interaction with dimeric PEAK scaffolds. Notably, relative to its monomeric form, as observed in SEC studies, dimeric CrkII^{FL} binds more tightly to all PEAKs (reflecting avidity of dimer:dimer interaction involving dual NSH3/PRM interfaces) (Fig. 8a). Additionally, whilst monomeric CrkII^{FL} can already bind PEAKs (via NSH3), the overall affinity of this interaction is markedly enhanced phosphorylation of PEAKs by Src, reflecting the combined effect of simultaneous SH2 and NSH3 engagement. In contrast, for Grb2, significant binding to PEAKs is only observed following Src phosphorylation (and only for PEAK3^{FL} and PEAK1^{IDR1}), confirming that direct Grb2 recruitment to PEAKs is primarily SH2-mediated and likely occurring via the consensus pY-SH2 motif we have identified and characterised.

In summary, we have now demonstrated the interaction of 14-3-3, CRKII, Grb2 with PEAK1/2/3 constructs and the results are consistent with those observed with peptides.

Modifications to manuscript: We have modified the main text by including the entire paragraph shaded above in the section “**Building a model for PEAK3/CrkII/Grb2 regulation via the tandem site 14-3-3 motif**” line XXX.

Figure 7 is now called **Figure 8** and two new panels have been created 8a and 8b, summarising new SPR data. See also Tables in supplementary section: Table Ext7 and Table Ext7 SPR data

The methodologies describing the purification of PEAK3-FL from *E. Coli* and SPR binding studies of PEAK proteins to full length CrkII/Grb2 and subdomains have been added in the online content.

2. The Grb2 SH2 part (Fig 2) seems a bit disconnected from the rest of the paper and is not entirely novel, as this interaction was previously characterized (see ref. 14). Therefore, it could be moved to SI.

We have decided to keep the molecular characterization of the conserved PEAK3^{Y24}/PEAK1^{Y1107} SH2 motif with Grb2 as part of the main text as it describes an important regulatory point for PEAK3 (and PEAK1). Specifically, we **have focused on demonstrating the importance of the residues outside the pY motif in conferring additional affinity and specificity** by evaluating by SPR whether the extended region of high sequence conservation in PEAK1/PEAK3 C-terminal to the SH2 motif could confer additional CrkII^{SH2}/Grb2^{SH2} binding affinity or selectivity. SPR data of longer peptides including the predicted helical structure in AlphaFold2 confirm that these extended regions of PEAK3 and PEAK1 do appear to confer some additional affinity and selectivity to the interactions.

Modifications to manuscript: we have modified the main text under the section ‘**The conserved PEAK3^{Y24}/PEAK1^{Y1107} SH2 motif is phosphorylated by Src and interacts with Grb2^{SH2}**’ as follows (beginning line 177):

In our orthologue multiple sequence alignment (MSA) data there is an extended region of high sequence conservation in PEAK3 and PEAK1 C-terminal to this SH2 motif (Fig. 2c-d). This corresponded to a predicted helical structure in AlphaFold2 (AF2)^{40,41} models of PEAK3 and PEAK1 (Fig. 2c-d). We were thus interested to evaluate whether this extended region could confer additional CrkII^{SH2}/Grb2^{SH2} binding affinity or selectivity. SPR studies were conducted with the shorter consensus pY SH2 motif phosphopeptide, or longer PEAK3²³⁻³⁸pY²⁴ and PEAK1¹¹⁰⁶⁻¹¹²¹pY¹¹⁰⁷ phosphopeptides to measure binding to immobilised full length Grb2 and CrkII (monomer and dimer) or individual SH2 domains. Together, these data confirm that these extended regions of PEAK3 and PEAK1 do appear to confer some additional affinity and selectivity to the interactions, with PEAK3²³⁻³⁸pY²⁴ exhibiting tighter binding affinity and selectivity towards Grb2^{SH2}/Grb2^{FL}, whereas for PEAK1¹¹⁰⁶⁻¹¹²¹pY¹¹⁰⁷ the additional region enables tighter binding to CrkII^{SH2}/CrkII^{FL}, but does not substantially alter binding to Grb2 (Fig. 2e).

Fig.2 has been modified, now showing in **b**, Zoom in structure of the PEAK SH2-pY peptide bound to the Grb2^{FL} dimer, with the PEAK phosphopeptide. **c,d**, PEAK3²³⁻³⁸ and PEAK1¹¹⁰⁶⁻¹¹²¹ regions alongside full MSA/Web Logo of orthologs for the corresponding sequence, showing a section of high sequence conservation adjacent to the TYSNL SH2 site. **e**, Tabulated SPR-determined mean steady state binding affinity values (K_D) for PEAK phosphotyrosine peptides (consensus pY-SH2 motif and extended peptides PEAK3²³⁻³⁸pY²⁴ and PEAK1¹¹⁰⁶⁻¹¹²¹-pY¹¹⁰⁷) binding to CrkII^{SH2}, full-length CrkII (CrkII^{FL}) monomer or dimer, Grb2^{SH2} or full length Grb2 (Grb2^{FL}). Also see Supplementary Information (Supplementary Table Ext1 and Table Ext1 SPR data for full tabulated data and sensorgrams). Also see relevant section for SPR interaction studies of full-length proteins, including Figure 8a, Supplementary information Table Ext 7 and Ext7 SPR data) which helps to further contextualize the Grb2 SH2 interaction in the broader context of the paper.

Extended data Fig. 2 has also been modified accordingly. Workflow to generate HADDOCK docking model of extended peptides has been removed and replaced by representative SPR sensorgrams.

Online method: see *SPR binding studies of PEAK pY phosphopeptides to CrkII/Grb2 and subdomains.*

3. Full results of ITC data need to be reported. Besides the K_d, N, deltaH and delta S need to be reported. All values (K_d, N, deltaH and delta S) should be reported for all independent replicates instead of only reporting the mean+/-SEM.

All ITC data can be found in Supplementary Information. Full details for all titrations are listed in Table Ext4A and the corresponding tab “Table Ext4 ITC data”; mean values from all titrations are collated in Table Ext4B.

Minor points:

1. Fig 3c/d and 4a: SDS-PAGE gels are not properly labeled. It is unclear which SEC fraction/elution volume was loaded in which lane. Furthermore, the molecular weight marker are not labeled.

We have now properly labeled the markers in Fig3c/d and 4a and have highlighted on the chromatogram which fractions have been loaded onto SDS page.

2. Fig 2a and line 156-158: MS data should be shown in SI. It is currently ‘hidden’ in source data without reference in the results.

It was not our intention to hide the MS data – we were following the journal guidelines for inclusion of raw data such as MS spectra. We have kept those in Source data but have provided a reference in the text line 158 where to find them.

Reviewer #2 (Remarks to the Author):

Review of Roy, et. al., 2022, “Structural insights into regulation of the PEAK3 pseudokinase scaffold by 14-3-3”

In this paper, Roy et. al biochemically and structurally characterize the interactions of the pseudokinases PEAK1, PEAK2, and PEAK3 with their binding partners CrkII, Grb2, and 14-3-3. The authors conducted a bioinformatic analysis to identify several sequence motifs in PEAK3 that mediate these interactions and solve crystal structures of a peptide derived from PEAK1/PEAK3 bound to the Grb2 SH2 domain, as well as phosphopeptides from all three PEAK family members bound to 14-3-3. This study also uses biochemical and biophysical studies to analyze the similarities and differences in the way each of the PEAK pseudokinases interacts with CrkII and 14-3-3, as well as the stoichiometry of these interactions and competition between these proteins for binding to PEAK3. Overall, this study reveals some new insights into the molecular basis for signaling by the PEAK pseudokinases, but **falls short of advancing the field enough to merit publication in this journal.**

We acknowledge that as a class, SH2 and 14-3-3 interactions have been extensively studied previously and that the characteristic binding motifs are well understood – this is not the key contribution of our work. However, the PEAK/SH2 and PEAK/14-3-3 crystal structures we have determined are important data to unambiguously demonstrate the interactions of PEAK3/1/2 interact with Grb2 and 14-3-3 at these sites according to the understood consensus motifs. Aside from this, as PEAK proteins are potential targets of interest in oncology, high resolution structures of these interactions are of interest to determine towards to inform possible future therapeutic targeting of PEAK protein-protein interactions with key signalling partners.

In terms of broader context to advance the field, we do believe that the paper makes significant contributions to advance the understanding PEAK-family pseudokinase biology specifically and methodologies for study of pseudokinase regulation in general.

PEAK3 is an intriguing dimeric pseudokinase scaffold of interest in oncology and while some interactors of PEAK3 have been identified, minimal biochemical analysis of these interactions has been reported to date. We provide an **extensive dataset to understand the bivalent and multivalent interactions of PEAK3 with key signalling partners, CrkII and Grb2, including important data to understand mechanisms of recruitment and the role of avidity in these interactions.**

In addition, perhaps the most important novel finding of our work is to **identify a new role for 14-3-3 proteins as an important regulator of PEAK3.** We biochemically unpick mechanisms for high affinity 14-3-3 recruitment and regulation of PEAK3 and demonstrate in cells the effects of 14-3-3 recruitment on PEAK3/CrkII/Grb2 signalling and cell motility. We feel these significant advances to the field are an excellent fit for Nature Communications and would be of interest to the readership of this journal.

While much effort likely went into the four crystal structures in this study, the data collection and refinement statistics for which are quite good, suggesting the structures are of high quality, unfortunately, these structures do not provide much new insight into the interactions with Grb2 SH2 or 14-3-3 beyond what would be expected based on the many previously published structures of complexes of SH2 domains bound to pTyr containing peptides or 14-3-3 proteins bound to pSer containing peptides. There are plenty of such structures already published (for example, >150 1433 – peptide complexes in the PDB, maybe lesser number of SH2domain-peptide complexes, but there are plenty).

1-Structure of PEAK3-FL bound to SH2 domain from CrkII or PEAK3-FL-bound to 1433 may reveal extended interactions that beyond what is predicted from domain / peptide analyses. There are several examples of these, so the more relevant context is missing here.

Please see comments above regarding SH2/14-3-3 crystal structures.

While we were conducting our studies, we became aware of Natalia Jura/ Klim Verba's work on the structure of PEAK3-FL in complex with 14-3-3. Our work provides a deep biochemical and biophysical analysis of the PEAK interactome that is essential for advancing our understanding PEAK signaling specificity. Our paper and Natalia Jura/ Klim Verba's paper are highly complementary and were co-submitted to Nature Communications so the context will not be missed when both papers are published back-to-back.

2-Moreover, the electron density in Fig 2d for the peptide in the Grb2 SH2 structure appears relatively weak, with only the phosphate and some of the backbone having strong density. The authors speculate that the predicted helix adjacent to the pTyr site is important for binding, but have no experimental data to support this. A crystal structure of a longer peptide including both the helix and pTyr site bound to Grb2 SH2 would elevate the work to some extent, corroborating their computational analysis, including experiments like testing whether mutagenesis of residues in the helix weakens binding.

We made several attempts to obtain a crystal structure of Grb2 SH2 with a longer peptide but were unsuccessful. **However, we have obtained supporting experimental data demonstrating that the residues adjacent to the pTyr site of PEAK3 and PEAK1 do appear to confer some additional affinity and selectivity to the interactions.** Please refer to our reply to reviewer 1, major point 2 for our full response and new data.

3- The observations that CrkII homodimerizes and that only homodimers, and not monomers, interact with the PEAK pseudokinases is very intriguing, but the physiological relevance of the dimer is unclear. It is interesting that only ~10% of the CrkII protein dimerizes. Since the authors did not find any differences between the monomer and dimer protein via MS, presumably the monomer and dimer exist in an equilibrium. It would strengthen the author's hypothesis if they reinjected the CrkII dimer complex or CrkII monomer complex onto

SEC to see if the CrkII dimer or monomer remain intact or sample then re-equilibrates into monomer:dimer as seen in the manuscript – of course, this experiment should be done with the same “amount” of the protein injected into SEC. The authors suggest that CrkII might be forming a domain swapped dimer, as has been observed in crystal structures of other SH2 domains.

We thank the reviewer for the suggestion. We have further tested the stability of the CrkII dimer by reinjecting purified CrkII dimer and CrkII monomers onto SEC to see if the CrkII dimer and monomer remain in their original conformational states. Experiments were conducted as followed:

Following purification of CrkII^{FL} dimer and CrkII^{FL} monomer, two 110 μ L samples of each of 50 μ M CrkII^{FL} dimer and CrkII^{FL} monomer (total 4 samples) were prepared and re-run them on SEC, S200 10/300 pre-equilibrated in 20 mM HEPES pH 7.5, 150 mM NaCl, 1 mM TCEP, on the day (reference samples) and after 45 hours incubation at 4°C.

Our data clearly indicates that both CrkII dimers and CrkII monomers exist as stable states even after 45 hours at 4°C and there is no evidence of a dynamic equilibrium. Therefore, we are confident that only CrkII dimers display a higher affinity for PEAK proteins, likely through an avidity effect (Extended data Fig. 3c).

Modifications to manuscript: We have modified the main text as followed by adding line 230 the following: “Further stability test conducted on purified CrkII^{FL} monomers and dimer indicates that both exist as stable states (Extended Data Fig. 3c)”.

Extended data Fig 3c. We have added the SEC stability data in Extended data Fig. 3c.

Online method. The following has been added:

To test the stability of purified of CrkII^{FL} dimer and CrkII^{FL} monomer, two 110 μ L samples of each of 50 μ M CrkII^{FL} dimer and CrkII^{FL} monomer were re-run on SEC, S200 10/300 pre-equilibrated in 20 mM HEPES pH 7.5, 150 mM NaCl, 1 mM TCEP, on the day and after 45 hours incubation at 4°C.

4-Based on structures of these other domain-swapped SH2 dimers, it might be possible to disrupt CrkII dimerization through mutagenesis. A monomerizing mutation would enable testing of the relevance of CrkII dimerization in cells, and establishing the physiological relevance of CrkII dimerization would increase the impact of this work.

While we agree that generating CrkII dimer mutants could add some physiological relevance to CrkII dimerisation in cells, this requires extensive new experiments beyond the scope of this paper. Mutations in CrkII may have multiple impacts in cells as it is an adaptor protein that plays a central role in signal transduction pathways integrating myriad signals and controlling tyrosine kinase signalling. Additionally, in signalling at focal adhesions, CrkII can be recruited via the SH2 domain to multiple adjacent phosphotyrosine substrate recruitment sites on p130cas/NEDD9. A SH2-mediated dimer of CrkII^{FL} is one possible example of clustered CrkII in cells, whereas another example may be two CrkII^{FL} monomers bound via the SH2 domains to adjacent sites on p130Cas. As we note in the manuscript, the potential for avidity in interactions of dimeric PEAKs with two adjacent free CrkII NSH3 domains is relevant to either of these possibilities and we exemplify it for the CrkII^{FL} SH2-mediated dimer.

Deconvoluting data in cells with CrkII dimer mutants would require a significant undertaking, the first step of which would be to identify residues mediating dimerisation. We have made efforts to crystallise the dimeric form of the CrkII SH2 domain. Dimeric CrkII SH2 was subjected to crystallisation experiments and well-diffracting crystals were obtained in multiple different conditions. The best of these resulted in a structure of the CrkII SH2 domain to 1.17Å. However, in all cases the obtained structure was of monomeric CrkII SH2 and was identical to a previously deposited structure of monomeric CrkII SH2 already in the PDB (pdb 5jn0, 1.68Å

resolution), including identical unit cell parameters and space group and identical compact crystal packing. Based on this crystal packing, we suspect that the propensity to crystallise in this monomeric form may make it highly challenging to obtain a structure of the dimeric form of the CrkII SH2 domain alone.

An alternative approach might be to make an educated guess on residues mediating dimerisation. Although there are a number of SH2-mediated dimers described in the literature (eg.Grb2 : <https://doi.org/10.1042/BCJ20210105>; doi: [10.2142/biophysico.16.0_80](https://doi.org/10.2142/biophysico.16.0_80)) the residues involved in dimerisation vary for different SH2 domains. In the absence of structural information to guide mutagenesis studies, this would be highly speculative and many varied point mutants would need to be cloned, expressed and purified to assess monomeric/dimeric state, with no guarantee that it is possible to achieve such a monomerising point mutant in practice, or perhaps whether a combination of mutations might be required.

Assuming a successful monomerising point mutant could be obtained, it would then be necessary to validate this functionally in a cellular context. This would require generation of CrkII- (and possibly CrkL) deficient cell lines to minimise the risk of endogenous CrkII/CrkL confounding interpretation of results. The overexpressed monomerising point mutants of CrkII would need to be validated to ensure these are folded and functional, before experiments could be then begin to examine a change in interaction with PEAK3.

Due to the extent of work to achieve these experiments in practice, it would amount to an entirely new line of study that we consider is beyond the scope of the present manuscript.

5-The authors also present biophysical data demonstrating that CrkII and 14-3-3 compete for binding to PEAK3, which has interesting implications for regulation of the PEAK3/CrkII interaction. While the biophysical data supporting this is strong, the cellular IP data is much weaker. For Extended Data Fig 4b, the authors do admit the difference in band intensity for pulldown of CrkII between the WT and S69A mutant is modest. This would be more convincing if there were data for a cellular assay measuring signaling downstream of CrkII that showed a difference when PEAK3 WT or S69A was expressed.

We have now added additional data to demonstrate the impact in cells of PEAK3/14-3-3/CrkII interaction on downstream signalling.

- a- As stated in the paper, we see with the S69A mutant defective for 14-3-3 binding (Fig 4b), a modest increase of CrkII binding (Extended Data Fig 4b). This is also accompanied also by a slight increase in Grb2 binding (data added in Extended Data Fig 4b).
- b- To further demonstrate the impact of PEAK3/14-3-3/CrkII interaction in cells, we expressed PEAK3^{FL}-WT and PEAK3^{FL}-S69A in MCF-10A cells and assessed PEAK3^{FL} tyrosine phosphorylation by immunoprecipitation and western blotting experiments using a pan pTyr antibody. **We found expression of PEAK3^{FL}-S69A mutant results in a significant increase of PEAK3 pTyr phosphorylation compared to the WT protein** (Fig. 7a). Our previous study demonstrated that mutation of Tyr24 abolished PEAK3 tyrosine phosphorylation and that Tyr24/Grb2 represents a key PEAK3 signaling axis (Science Signalling paper). Taken together, these data indicate that abolishing 14-3-3 binding, enhances CrkII binding and pY24/Grb2 mediated downstream signalling.
- c- These data prompt us to assess the impact of S69A mutation on cell motility and demonstrated that **PEAK3^{FL}-S69A displays a significant increase in cell motility indicating that 14-3-3 may acts as a break for pY24/Grb2 motility signalling** (Fig. 7b).

Modifications to manuscript: We have modified the main text as followed:

Line 387: this is also accompanied also by a slight increase in Grb2 binding - Extended Data Fig 4b modified accordingly and corresponding uncropped gels added in source data.

Line 397: To further demonstrate the impact of PEAK3/14-3-3/CrkII interaction in cells, we expressed PEAK3^{FL}-WT and PEAK3^{FL}-S69A in MCF-10A cells and assessed PEAK3^{FL} tyrosine phosphorylation by immunoprecipitation and western blotting experiments using a pan pTyr antibody. We found expression of PEAK3^{FL}-S69A mutant results in a significant increase of PEAK3 pTyr phosphorylation compared to the WT

protein (Fig. 7a). Our previous study demonstrated that mutation of Y²⁴ abolished PEAK3 tyrosine phosphorylation and that the Y²⁴/Grb2 interface represents a key PEAK3 signaling axis¹⁴. Taken together, these data indicate that abolishing 14-3-3 binding, enhances CrkII binding and pY²⁴/Grb2 mediated downstream signalling. As Grb2 signaling has a crucial role in regulating actin-based cell motility^{25,53}, we next directly assessed the impact of PEAK3 S69A mutation in this context. Strikingly, cell motility in PEAK3^{FL}-S69A expressing cells is significantly increased, supporting the conclusion that 14-3-3 represents a key direct regulator of PEAK3 function and may thereby provide a brake on PEAK3 pY²⁴/Grb2 motility signalling (Fig. 7b).

Figures: New Figure 7 created: S69A mutant increases PEAK3 p-Tyr and cell motility.

Online method: Reagents and protocols have been added under “cellular studies of PEAK interaction”

We believe these additional cell-based experiments support our in depth biochemical / biophysical characterisation of PEAK interactions.

Other comments:

1) For the kinase reaction in which Src was shown to phosphorylate SH2 motif on PEAK3, was the reaction specific only to that specific Tyrosines mentioned or was it heterogenous?

The kinase reaction in which Src was shown to phosphorylate the SH2 motif on PEAK3 is specific to that tyrosine (Y24), no other tyrosine phosphorylation was observed.

2) Why not utilize the extended peptide for the SH2-domain binding region (the one with secondary structure predicted from Alpha-fold) in the crystallization trials?

We have attempted to get a crystal structure of the extended peptide for the SH2-domain binding region but were unsuccessful. However, we have obtained supporting experimental data demonstrating that the residues adjacent to the pTyr site of PEAK3 and PEAK1 do appear to confer some additional affinity and selectivity to the interactions. **Refer to reviewer 1, major point 2 for full response with additional data.**

3) Page 6, towards the end there is a lot of speculation without much data and distracts from the rest of the study.

We have removed all speculation and instead added additional binding data with longer peptides (see Reviewer 1 – major point 2).

4) Does CrkII NSH3 domain bind to the tandem peptide containing the PRM and 1433 motif in presence of 1433 in the context of full length PEAK3 or the peptide itself?

The ITC studies we conduct with the tandem PEAK3 peptide confirm that it is possible for CrkII NSH3 domain to bind to the pS69-phosphorylated PEAK3 tandem peptide even when 14-3-3 is bound, however the affinity of this interaction is reduced relative to the same peptide in the absence of 14-3-3 (resulting in negative cooperativity) (Fig.6).

We also show for a full-length PEAK3 dimer bound to 14-3-3 that this reduction in binding is sufficient to prevent dimeric CrkII^{FL} from stably interacting with the PEAK3:14-3-3 complex on SEC (Fig 4d), whereas dimeric PEAK2 in the absence of bound 14-3-3 forms a stable complex with dimeric CrkII^{FL} (Fig 3c).

In addition, in response to reviewers comments we have conducted additional SPR binding experiments for CrkII^{FL} monomer and dimer binding to PEAK3^{FL} (purified as 14-3-3 complex from insect cells) as well as PEAK3^{FL} from *E.coli* (refer Reviewer 1 major comment 1 above) and both PEAK1^{IDR1} and PEAK2^{IDR1}, each of which did not co-purify with 14-3-3. In the case of the immobilised PEAK3^{FL}/14-3-3 complex purified from insect cells, we observed that insect cell-derived 14-3-3 dissociated from immobilised PEAK3^{FL} under the

constant flow conditions on the SPR sensor chip. This was evident in the robust reversible binding/dissociation of additional recombinant 14-3-3 γ , to a level (based on achieved response units) consistent with saturation of this pS69 binding site (Supplementary information, Table Ext7 SPR data). Based on this it appears that whilst it carries the pS69-phosphorylation mark, immobilised PEAK3^{FL} from insect cells has essentially lost bound 14-3-3 when the binding experiments with CrkII^{FL} are conducted. Consistent our SEC studies, we find by SPR that dimeric CrkII^{FL} binds tighter than the monomeric form for all PEAKs (Fig 8a), including PEAK3^{FL}, whether expressed in insect cells or bacterial cells.

Unfortunately, as competition experiments to measure CrkII^{FL} binding to 14-3-3-bound PEAK3 by SPR are more technically challenging than by ITC or SEC, we have not been able to undertake these in addition to the SEC studies with full-length protein and ITC studies using purified peptides. ITC competition studies with PEAK3^{FL}/14-3-3 (from insect cells) or PEAK3^{FL} (from *E.coli*) were not possible due to the amount of PEAK3 protein that would be required.

5) Page 9 – PEAK3:14-3-3 for a stable dimer-dimer heterocomplex – this is a well written section and very relevant and interesting data. The authors should consider obtaining a structure of this complex, with using cryoEM or by co-overexpressing PEAK3-FL with a human 14-3-3 isoform and perform tandem purification with different affinity tags and isolate a homogenous complex for crystallography, potentially.

While we were attempting to obtain a structure of PEAK3-FL in complex with 14-3-3, we were made aware by Natalia Jura and Kliment Verba that they had solved PEAK3-FL/14-3-3 by cryo-EM. This prompted us to work collaboratively with Natalia and Kliment to produce complementary papers and co-submit our studies.

This article can be found at: bioRxiv 2022.09.01.506268; doi: <https://doi.org/10.1101/2022.09.01.506268>
Structural insights into regulation of the PEAK3 pseudokinase scaffold by 14-3-3 ; Hayarpi Torosyan, Michael D. Paul, Antoine Forget, Megan Lo, Devan Diwanji, Krzysztof Pawłowski, Nevan J. Krogan, Natalia Jura, Kliment A. Verba

Reviewer #3 (Remarks to the Author):

The study attempts to build a more comprehensive model of complexes formed by the PEAK proteins, and provides structural and biophysical analyses which allow extrapolation of how these complexes occur. The work clearly enhances the understanding of PEAK proteins and provides new information on their interactions, but there are a number of areas in which I believe the study could be made more robust. These are discussed below.

1-The crystallographic analysis of Grb2 is clear, and although the characteristic turn in the peptide is obviously revealed they may consider showing the refined 2Fo-Fc/Fo-Fc maps in addition to their unbiased Fo-Fc omit map of the bound PEAK peptide bound. In contrast, the HADDOCK modeling discussed on page 6 concludes that a new conserved surface is created, but this is based on modeling a model's interaction with their structure. This HADDOCK analysis should be removed as too speculative unless it can be supported by experimental evidence such as ITC to demonstrate the extended predicted interaction enhances selectivity for Grb2 over CrkII (i.e. better than the 2.6 μ M vs 7.8 μ M Kds which they previously published for the short 7-mer peptide).

Refined 2Fo-Fc map is now shown (in Fig.2) in addition to the unbiased Fo-Fc omit map (Extended Data Fig.2c)

We have removed the HADDOCK modeling discussed on page 6. Instead, we have focused on obtaining supporting experimental data for longer peptides incorporating the residues adjacent to the pTyr site of PEAK3 and PEAK1. This data demonstrates that these additional residues appear to confer some additional affinity and selectivity to the interactions. **Refer to reviewer 1, major point 2 for full response with additional data.**

2-The SH3 analysis of CrkII with PEAK provides a clear description of this interaction, and it is indeed striking that the dimeric CrkII-FL elutes with the dimeric PEAK by SEC. For completeness, they may wish to include the SEC-MALS for full-length CrkII in Extended Fig 3e and an analysis of the affinity of the Grb2 SH3 domain for the PxxP motif. They may also consider the possibility that the tyrosine phosphorylation at the SH2 motif is proximal (~30 residues) to the PxxP motif, so a single CrkII (or indeed Grb2) protein should be able to engage both the pY and PxxP motifs with potentially significantly tighter binding and avidity than the observed dimer-dimer SH3-mediated complex. This would also impact both their pY-SH2 analysis and 14-3-3 cooperativity analysis.

The SEC-MALS of full-length CrkII has been included in Fig 3e.

We agree with the reviewer's pertinent comment regarding the potential for simultaneous engagement of both pY and PxxP motifs that may confer additional avidity to the interaction.

Such experiments involving multivalent interactions are inherently more challenging to address as compared to a 1:1 interaction. However, we have made efforts to address this by conducting an additional comprehensive SPR study with full-length PEAK3 and full-length CrkII/Grb2 forms (as well as individual sub-domains) (Fig 8a), both in the presence and absence of Src phosphorylation of PEAKs.

The data confirm binding of CrkII and Grb2 SH2 only to Src-phosphorylated PEAK3^{FL} (insect cell or bacterially expressed), whereas CrkII NSH3 binds to PEAK3^{FL} (both forms) with very similar affinity irrespective of Src phosphorylation.

These data recapitulate the striking increase in binding of CrkII^{FL} dimer to PEAK1/2 relative to the monomer (NSH3 mediated avidity effect) in the absence of Src phosphorylation. This corroborates observations by SEC and extends to PEAK3^{FL}. We also observe that CrkII^{FL} monomer shows enhanced binding to PEAKs following Src phosphorylation, which we attribute to a combined NSH3- and SH2-mediated avidity, as per the possibility alluded to by the reviewer.

As compared to CrkII^{FL} monomer, the effect of Src phosphorylation on binding to PEAK3/PEAK1 is even more pronounced for Grb2^{FL}, in which the SH2-mediated interaction appears to constitute the dominant direct interaction site. It is possible that Grb2 SH3-1 and SH3-2 have some weak binding for PxxP motifs on PEAKs, but based on the binding data for Grb2^{FL} this contribution is significantly weaker ($K_D > 50 \mu\text{M}$) than that of the CrkII^{NSH3} with PxxP motifs on PEAKs ($K_D \sim 1\text{-}5 \mu\text{M}$).

We have addressed this in the main text by modifying as follows:

Line 416:

Building a model for PEAK3/CrkII/Grb2 regulation via the tandem site 14-3-3 motif

To build a comprehensive model of PEAK/CrkII/Grb2 interactions and extend our studies with isolated PEAK peptides, we next undertook SPR binding studies using immobilised biotinylated full length or N-terminally truncated recombinant PEAKs (insect cell expressed human PEAK1^{IDR1}, PEAK2^{IDR1}, PEAK3^{FL}-14-3-3 complex and bacterially expressed PEAK3^{FL}) to measure the affinity of interactions with full-length interactors and relevant sub-domains (14-3-3 γ , CrkII^{FL} monomer or dimer, Grb2^{FL}), both with or without prior phosphorylation of PEAKs by Src kinase (Fig. 8a).

These SPR studies confirmed the key interactions previously mapped using individual peptides. Notably, whilst insect cell expressed PEAK3^{FL} (S69 phosphorylated) was purified and immobilised as a complex with insect derived 14-3-3 (refer experimental methods), it showed strong and reversible binding to injected recombinant human 14-3-3 γ , confirming that endogenous insect 14-3-3 had dissociated from PEAK3^{FL} under the continuous flow conditions in SPR experiment. Consistent with PEAK phosphorylation state and measured affinity of individual tandem peptides (Extended Fig. 5c), only insect cell expressed human PEAK3^{FL} (and no other PEAKs tested, including human PEAK3^{FL} purified from *E.coli*) bound strongly to 14-3-3 γ . Additionally, phosphorylation of PEAKs by Src was required Grb2^{SH2} and CrkII^{SH2} to bind, with Grb2^{SH2} demonstrating tightest binding to PEAK3^{FL}, whilst

CrkII^{SH2} bound most tightly to PEAK1^{IDR1} (Fig 8b) consistent with SPR results using individual pY phosphopeptides from the SH2 motif (Fig. 2e). As for studies using PEAK tandem site proline rich motif peptides, the affinity of CrkII^{NSH3} towards PEAK dimers was found to be relatively similar across all PEAKs (K_D in low micromolar range) irrespective of Src phosphorylation state, reinforcing the understanding of the NSH3/PRM interaction as a primary PEAK-CrkII recruitment site (Figures?). The data also reveal contributions of multiple interaction domains to avidity in interaction with dimeric PEAK scaffolds. Notably, relative to its monomeric form, as observed in SEC studies, dimeric CrkII^{FL} binds more tightly to all PEAKs (reflecting avidity of dimer:dimer interaction involving dual NSH3/PRM interfaces) (Fig. 8a). Additionally, whilst monomeric CrkII^{FL} can already bind PEAKs (via NSH3), the overall affinity of this interaction is markedly enhanced phosphorylation of PEAKs by Src, reflecting the combined effect of simultaneous SH2 and NSH3 engagement. In contrast, for Grb2, significant binding to PEAKs is only observed following Src phosphorylation (and only for PEAK3^{FL} and PEAK1^{IDR1}), confirming that direct Grb2 recruitment to PEAKs is primarily SH2-mediated and likely occurring via the consensus pY-SH2 motif we have identified and characterised.

3-The discovery of the 14-3-3 binding is indeed interesting, and another nice example of ‘fishing’ out 14-3-3 from baculovirus. The crystal structures of 14-3-3/PEAK should be supported by unbiased Fo-Fc omit maps and refined 2Fo-Fc and Fo-Fc maps for all three bound peptides, and Table 1 should present the B-factors for each chain (14-3-3 and PEAK peptides) and should clearly state the peptide residues which are built in each peptide chain.

Unbiased Fo-Fc omit maps and refined 2Fo-Fc and Fo-Fc maps for all 3 peptides (and each of the four chains for the PEAK3 tandem-pS69/14-3-3 ϵ structure) have been added and can now be found in Extended Data Fig.5.

B-factors for each chain and peptide (and protein) residues built for each chain are now stated in **Table 1**.

4-SPR data and traces for the tandem phosphopeptides binding to 14-3-3 isoforms should be presented in an extended figure (currently these data are presented as a heatmap).

All SPR data and traces were provided and could be found in Supplementary Information. Considering the large amount of data, we leave it to the journal to decide where those should be included, in Extended Data or Supplementary Information.

SPR data and traces for the additional experiments conducted following review have also been added to the Supplementary Information alongside the original SPR data.

5-Additionally, the gaps in the sequence alignment (Figure 1b) for PEAK1 and PEAK3 are in the wrong places; RTQSLP, RANTEP and RAASSP should all align directly above one another.

We have redone the sequence alignment and modified Fig. 1b accordingly.

6-The titrations presented in Figure 6 are a nice series of experiments. They might briefly discuss why they used 14-3-3 gamma rather than 14-3-3 epsilon which they used for the crystallography. They also should include an extended table of their ITC parameters (n , dS , dH etc) for each run. Their conclusion that there is negative cooperativity between 14-3-3 binding to PEAK3 and CrkII binding would be better supported by an SEC analysis of S69A PEAK3 interaction with dimeric CrkII-FL (as per Figure 4d), where based on their data the S69A mutant would be expected to interact with CrkII dimer in a manner similar to PEAK1 and PEAK2.

Initially, both ITC experiments and crystallisation studies were conducted primarily with 14-3-3 γ , which constituted a high affinity 14-3-3 interactor with the PEAK3 tandem pS69 site. However, although large and reproducible crystals were obtained with 14-3-3 γ and the PEAK3 tandem pS69 peptide, these consistently diffracted relatively poorly ($>3 \text{ \AA}$ resolution) and were unable to be further optimised. Possibly a modified 14-3-3 γ construct may improve resolution, however this was not pursued. In parallel, crystallisation had been

initiated with 14-3-3ε and well-diffracting crystals were obtained for this complex with PEAK3 tandem pS69 peptide.

The extended data with ITC experiments has been modified to include both tabulated data and individual traces for each individual ITC titration, rather than averaged data (refer Supplementary Information, Table Ext5A and Table Ext5 ITC data).

We have tried to express and purify S69A PEAK3 mutant, however this protein is not soluble in our hands and could not be isolated in sufficient quantities. To circumvent this, we have attempted to purify PEAK3^{FL} from *E. coli* and succeeded in obtaining a small amount of pure PEAK3^{FL}, unphosphorylated and not complexed with 14-3-3. Refer Reviewer 1, major comment 1.

Whilst this *E.coli* purified PEAK3^{FL} was obtained in insufficient quantity for SEC studies, we have been able to successfully immobilise it for SPR studies to measure binding to CrkII^{FL} monomer and dimer, alongside PEAK1^{IDR1} and PEAK2^{IDR1} (which also did not co-purify with 14-3-3). The SPR studies confirm that, in the absence of bound 14-3-3 and Src phosphorylation, all PEAKs bind strongly to CrkII^{FL} dimer in a similar manner and bind relatively more weakly to CrkII^{FL} monomer (Fig.8a), as would be anticipated based on SEC studies and SPR studies using isolated PRM peptides.

Minor –

Line 263 – Extended Data Fig 4a refers to wrong panel (should be Fig 4a)

Corrected

REVIEWERS' COMMENTS

Reviewer #1 (Remarks to the Author):

The authors have considerably modified the manuscript and added a significant amount of important experimental data to the manuscript that further strengthens the manuscript. All comments/suggestions that I raised during the initial review were fully addressed.

Reviewer #2 (Remarks to the Author):

The authors have done a good job in their thorough responses to my comments and concerns, and I agree the work is complementary to the work on the PEAK3:1433 complex by Jura/Verba labs. I support the acceptance of this manuscript in its current form and its publication.

Reviewer #3 (Remarks to the Author):

The authors have completed a comprehensive response to the comments. I have no further concerns.